



# Characterization of Ice Features in the Southwest Greenland Ablation Zone Using Multi-Modal SAR Data

Sara-Patricia Schlenk[1,2], Georg Fischer[1], Matteo Pardini[1], and Irena Hajnsek[1,3]

[1]German Aerospace Center (DLR) - Microwaves and Radar Institute (HR)
[2]Friedrich-Alexander University Erlangen-Nuremberg (FAU) - Institute of Microwaves and Photonics
[3]Swiss Federal Institute of Technology Zurich (ETH) - Institute of Environmental Engineering

**Correspondence:** Sara-Patricia Schlenk (patricia.schlenk@dlr.de)

**Abstract.** This study investigates ice features of unknown glaciological origin in the ablation zone of southwest Greenland, focusing on the land-terminating Russell Glacier. Using data from an experimental airborne SAR (Synthetic Aperture Radar) campaign of the German Aerospace Center (DLR), the research employs a range of advanced techniques, including SAR polarimetry, interferometry, tomography, and modeling, to characterize these features. The analysis reveals that in low-backscatter

areas, surface scattering is dominant with no correlation to topography or surface characteristics. In contrast, surrounding high-backscatter areas are characterized by volume scattering and the presence of subsurface scattering structures. A significant aspect of this study involves comparing the observed ice features with known glaciological phenomena in the ablation zone. However, the combined findings, along with the temporal stability of these features, as seen through annual SAR backscatter analysis, complicate a straightforward glaciological explanation. A first theory involves the presence of a weathering crust

causing these low-backscatter features due to residual liquid water content. These findings could improve our understanding of surface and subsurface processes in the ablation zone, contributing to better mass balance assessments.

## 1 Introduction

The ablation zone of the Greenland Ice Sheet (GrIS) is one of its most dynamic regions, where ice melt drives significant meltwater production, contributing to global sea-level rise and reflecting the impacts of climate change (Box et al., 2012;

Fettweis et al., 2013). Situated at the lowest elevations of the GrIS, this zone experiences continuous ice loss that exceeds accumulation. Its bare ice, minimal snow cover, and prominent crevasses make it particularly vulnerable. Understanding these dynamic processes is essential for assessing the overall GrIS mass balance and the broader implications for global climate systems (Mote, 2007; Paterson, 1994). In this context, monitoring variations in ice properties within the ablation zone is crucial. Our observations revealed a puzzling spatial pattern of low-backscatter areas in Synthetic Aperture Radar (SAR)

data, suggesting corresponding variations in the ice's dielectric or structural properties. However, no preliminary glaciological explanation matched the SAR observations, rendering their investigation important to potentially enhance our understanding of surface and subsurface properties in the ablation zone.

SAR has become a valuable tool for studying and monitoring glaciers and ice sheets. Currently, monitoring of the GrIS relies heavily on space-borne X-band (8-12 GHz) and C-band (4-8 GHz) systems, such as Sentinel-1 and TanDEM-X, possibly in



combination with optical remote sensing data, to create e.g. Digital Elevation Models (DEM) (Krieger et al., 2007), observe
flow velocities (Joughin et al., 2017) and to classify glacier zones based on SAR backscatter (Paterson, 1994; Braun et al.,
2000). Overall, these higher frequency bands (X- and C-band) primarily capture surface characteristics. In contrast, lower
frequency bands such as L-band (1-2 GHz) and P-band (0.3-1 GHz) have the ability to penetrate deeper into the ice and can
therefore capture subsurface characteristics or internal structures of glaciers (Dall et al., 2001). However, while we have access
to space-borne data at these frequencies from systems like ALOS, their temporal and spatial coverage in the GrIS is limited
(Rosenqvist et al., 2007). Only through the analysis of low-frequency data acquired during airborne SAR campaigns have
aforementioned low-backscatter areas been identified (Banda et al., 2016; Parrella et al., 2021). Moreover, these areas remain
almost undetectable in both optical imagery and higher-frequency SAR data, making them fascinating features to investigate
in a multi-frequency analysis.

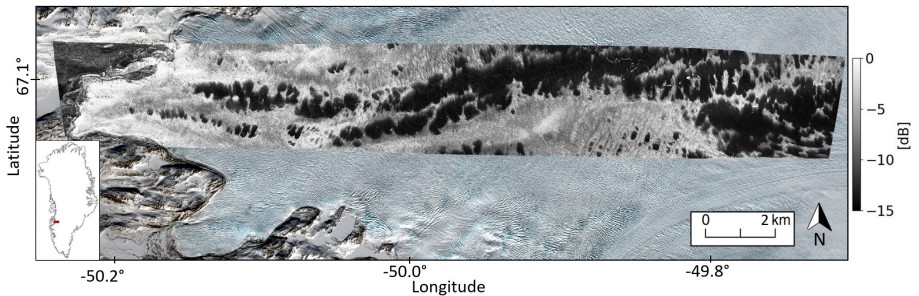

**Figure 1.** Radar-dark (low-backscatter areas) and radar-bright (high-backscatter areas) features in the ablation zone of southwest Greenland,
observed in airborne SAR at P-band with HV polarization (shown here at the Russell Glacier). In the background, a Sentinel-2 optical image
provides context for the surrounding area. In the lower left corner, a Greenland map highlights the test site with a red marker.

In this study, we refer to these low-backscatter areas on the glacier as radar-dark features, which are often surrounded by
high-backscatter areas, called radar-bright features. Our analysis shows that these radar-dark features exhibit distinct spatial
characteristics, typically appearing as round or oval shapes, measuring $100\,\mathrm{m}$ to $500\,\mathrm{m}$ in size. These features often form
interconnected patterns that contrast sharply with radar-bright features, creating a clear visual distinction across a section of
the Russell Glacier's ablation zone, as observed in airborne SAR data at HV polarization and P-band frequency (Fig. 1).
Low-frequency, space-borne SAR data further revealed that these radar-dark and radar-bright features represent a large-scale
phenomenon across the ablation zone of southwest Greenland, particularly in low-velocity, land-terminating glaciers. As this
study primarily relies on remote sensing data, and lacks ground measurements, the investigation of the unknown glaciological
origins of these radar-dark and radar-bright features is based on an analysis of their scattering processes and structures in
comparison with known glaciological phenomena.
To analyze the scattering processes of these ice features, we apply advanced SAR techniques: Polarimetric SAR (PolSAR),
Interferometric SAR (InSAR), and Tomographic SAR (TomoSAR). PolSAR differentiates scattering mechanisms by distin-
guishing surface, volume, and dihedral scattering (Cloude and Pottier, 1997). It is used in ice sheet monitoring to detect changes





in ice properties, such as surface roughness and volume scattering caused by snow or ice layers (Rott and Davis, 1993; Parrella et al., 2021). InSAR provides critical insights into the vertical structure of ice sheets. Key parameters such as coherence and

phase center height are crucial for interpreting the scattering processes of the study area (Rignot et al., 2001). Volumetric decorrelation, a major contributor to total coherence loss, can be linked to the vertical distribution of scatterers and helps to assess the subsurface structure (Bamler and Hartl, 1998; Fischer et al., 2020). TomoSAR provides 3D imaging capabilities (Reigber and Moreira, 2000), enabling detailed exploration of the ice subsurface, effectively mapping internal layers and detecting complex subsurface structures at different heights (Tebaldini et al., 2016; Banda et al., 2016). Moreover, conducting a temporal analysis

is essential for determining the origin of these ice features. We can observe annual variations through ALOS backscatter data (Bolon et al., 2007; Ruan et al., 2012), enabling us to monitor the formation and evolution of radar-dark features, providing critical insights into whether these features represent stable structures or transient phenomena.

After determining the scattering processes and temporal changes, we aim to gain a glaciological understanding of radar-dark and radar-bright features observed in our test site, the Russell Glacier's ablation zone. To achieve this, we compare these ice

features with known surface and subsurface processes, related to glaciological phenomena commonly found in the ablation zone. Considered surface phenomena include drained and refrozen supraglacial lakes (Hu et al., 2021; Yang et al., 2021), as well as physical impurities, such as the so-called "dark zone" and cryoconite deposits on the glacier surface (Wientjes and Oerlemans, 2010; Ryan et al., 2018). The subsurface phenomena investigated include seasonal temperature variations and englacial water bodies such as water pockets and channels (Catania et al., 2008; Lampkin and VanderBerg, 2013). We also

consider the weathering crust, a near-surface subsurface phenomenon, as observed at Russell Glacier by Cooper et al. (2018). All these phenomena may reduce backscatter through absorption, attenuation, or reflection, and could potentially be related to the radar-dark features. Moreover, by focusing our investigation on data collected during stable, frozen conditions, we aim to minimize the influence of active melt and better isolate the glaciological processes responsible for radar-dark features in SAR imagery.

The paper is organized as follows: Sect. 2 describes the test site and data. Sect. 3 outlines the methods, focusing on the SAR techniques applied. Sect. 4 presents the investigation of radar-dark and radar-bright features. Sect. 5 provides a discussion of the results in relation to glaciological processes, and Sect. 6 concludes with a summary of findings and potential directions for future research.

## 2   Test Site and Data

This study examines the ablation zone of southwest Greenland, with a focus on the Russell Glacier, also known as K-Transect due to the availability of NASA's Operation IceBridge data (MacGregor et al., 2021; Studinger et al., 2022) and long-term records from 'Programme for Monitoring of the Greenland Ice Sheet' (PROMICE) weather stations (Fausto et al., 2019). Located at approximately $67.1\,°N$ and $50.0\,°W$, the analysed area spans elevations from $200\,m$ to $860\,m$ above sea level (asl) for about $26\,km$ east-west and $3\,km$ north-south (Fig. 1). The average glacier flow velocity at the eastern edge of the test site

is approximately $100\,m\,y^{-1}$ (ENVEO, 2024), which decreases towards the terminus, as typical for land-terminating glaciers





in this region (Nagler et al., 2015; van de Wal et al., 2015). Ablation rates vary annually between $2.5\,\mathrm{m\,y^{-1}}$ to $5.6\,\mathrm{m\,y^{-1}}$ (ENVEO, 2023), although the terminus position has remained stable despite these fluctuations.

## 2.1 Data

### 2.1.1 SAR Data: The ARCTIC15 Campaign

During the ACTIC15 airborne campaign led by the German Aerospace Center (DLR) and the Swiss Federal Institute of Technology in Zurich (ETH), fully polarimetric and tomographic SAR acquisitions were performed using DLR's F-SAR system over the selected test sites at X-, L-, and P-band frequencies (Horn et al., 2008). Table 1 summarizes the acquisition types, look directions, frequencies and resolution for each flight. Fig. 2 illustrates the typical SAR side-looking geometry.

**Table 1.** Overview of flights, and their parameters, conducted during the ARCTIC15 campaign over the Russell glacier

| Acquisition date | Acquisition type | Look direction | Freq. band | Azimuth res. | Range res. |
|---|---|---|---|---|---|
| 05.04.2015 | Tomographic | Side-looking | X: 9.6 GHz | 0.5 m | 0.5 m |
| 05.04.2015 | Tomographic | Side-looking | L: 1.3 GHz | 0.6 m | 1.3 m |
| 12.05.2015 | Tomographic | Side-looking | P: 0.44 GHz | 1.0 m | 3.8 m |
| 15.05.2015 | Sounder | Nadir-looking | P: 0.44 GHz | | |

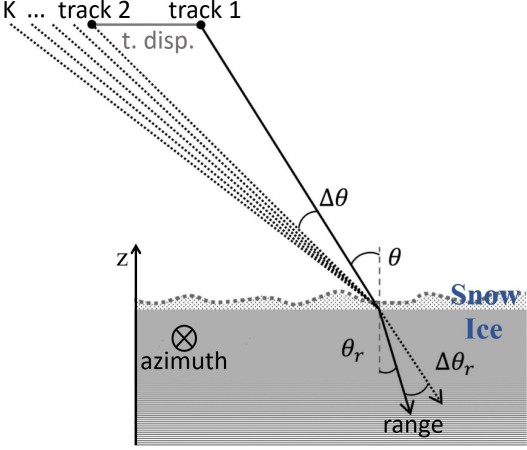

**Figure 2.** SAR side-looking geometry with multiple acquisitions from horizontally displaced tracks at the same height.

For each resolution cell within the covered area, after interacting with the ice elements, the transmitted electromagnetic
pulses are scattered back to the radar along the range direction, which forms an incidence angle $\theta$ with the surface normal. For the considered F-SAR acquisitions $\theta$ varies between $25°$ to $55°$, providing a swath width of around $3000\,\mathrm{m}$ for a flight height of $3000\,\mathrm{m}$. The azimuth direction, although not shown in Fig. 2, is perpendicular to the range-height plane and aligns with the



radar's flight path. As the SAR pulses penetrate into the ice, the incidence angle changes due to refraction from $\theta$ to $\theta_r$, which is dependent on the permittivity ($\varepsilon_r = 3.15$) and must be accounted for during data analysis.

In the ARCTIC15 TomoSAR acquistions, multiple tracks, horizontally displaced in a non-uniform way at the same flight height, were flown, as depicted in Fig. 2. The realized displacements provide sensitivity to the distribution of scatterers in the direction perpendicular to the range direction and therefore along the height $z$, enabling InSAR height estimations and TomoSAR imaging. The vertical wavenumber $\kappa_z$ describes the interferometric sensitivity to height. It is defined as (Papathanassiou and Cloude, 2001; Reigber and Moreira, 2000; Bamler and Hartl, 1998):

$$\kappa_z = \frac{4\pi}{\lambda} \frac{\Delta\theta}{\sin\theta} \tag{1}$$

where $\lambda$ is the radar wavelength, $\Delta\theta$ is the difference in incidence angles between acquisitions, and $\theta$ is the incidence angle. For subsurface characterization of ice, $\kappa_{zVol}$ accounts for refraction and permittivity effects (Sharma et al., 2013):

$$\kappa_{zVol} = \frac{4\pi\sqrt{\epsilon_r}}{\lambda} \frac{\Delta\theta_r}{\sin\theta_r} \tag{2}$$

where $\varepsilon_r$ is the relative permittivity of ice, $\theta_r$ is the refracted incidence angle, and $\Delta\theta_r$ is the difference in refracted angles.

The Rayleigh limit for the vertical resolution ($\rho_z$) of TomoSAR reconstructions depends on the maximum $\kappa_{zVol}$ across all the tracks (Reigber and Moreira, 2000):

$$\rho_z = \frac{2\pi}{\max\kappa_{zVol}} \tag{3}$$

Table 2 summarizes the tomographic parameters at the Russell glacier, at L- and P-band, including the number of tracks, displacements, and vertical resolution. Regardless of the frequency used, the minimum track displacement (t. disp. in Fig. 2)

was set at $5\,\mathrm{m}$ because of limitations in the flight accuracy. As $\kappa_{zVol}$ scales with the wavelength, the displacements at P-band were adjusted to provide the same tomographic performance as at L-band (including vertical resolution). Each TomoSAR acquisition was conducted within two hours under stable, frozen conditions with no variability from surface melt or internal structural changes. The slow glacial movement (less than $2.7\,\mathrm{m}$ during the campaign) at the Russell glacier ensured consistent ice properties across flights.

**Table 2.** Summary of acquisition parameters at the Russell glacier.

| Frequency band | Tracks | Track disp. [m] | $\kappa_{zVol}$ | $\rho_z$ |
|:---:|:---:|:---:|:---:|:---:|
| L | 7 | -35, -30, -20, 0, 20, 40, 55 | 0.13 to 1.69 | $3.72\,\mathrm{m}$ |
| P | 8 | -105, -90, -75, -65, 0, 60, 120, 165 | 0.07 to 1.50 | $4.20\,\mathrm{m}$ |

During one F-SAR flight over the Russel glacier, the P-band radar was operated as a sounder (i.e. nadir-looking along the vertical direction) collecting data along a transect in full polarimetric mode. Processing of this sounder data involves coherent summation of successive pulses and fully focused SAR processing to enhance along-track (azimuth) resolution. Motion compensation is adapted to the nadir-looking geometry (Scheiber et al., 2008). In order to avoid excessive across-track





(range) clutter and enable the sounder profile to accurately isolate reflections from shallow subsurface layers we selected the
cross-polarisation for our comparison.

It is worth remarking at this point that both TomoSAR's side-looking and sounder nadir-looking modes offer distinct sub-surface imaging capabilities. TomoSAR provides vertical backscattering profiles with penetration down to maximum $-80\,\mathrm{m}$ in P-band at the Russell glacier. In contrast, the sounder vertical propagation is subject to a lower attenuation than the side-looking slanted one, and penetration in ice increases to depths of several hundred meters.

### 2.1.2 Additional Datasets

Additional optical and space-borne SAR datasets have been considered to comprehensively analyze the test site at the Russell glacier (Table 3). Optical data includes high-resolution optical imagery from Sentinel-2, providing a way to infer large-scale surface characteristics (Drusch et al., 2012). Additionally, an orthophoto obtained from NASA's Operation IceBridge (MacGregor et al., 2021), an airborne mission delivering detailed, small-scale imagery, allows for detailed examination of surface 130 roughness and topography. Sampled height measurements have been provided by Airborne Topographic Mapper (ATM) lidar instrument from the NASA IceBridge campaign, enabling precise elevation assessments (Studinger et al., 2022). Finally, L-band SAR ALOS-2 data acquired between 2017 and 2024 have been used to detect annual changes of the ice features (JAXA, 2024).

**Table 3.** Supplementary SAR and optical data utilized at the Russell glacier

| Data type | Data | Acquisition dates | Spatial res. |
|---|---|---|---|
| Optical | Sentinel-2 | 04.05.2017, 21.07.2017 | $10\,\mathrm{m} \times 10\,\mathrm{m}$ |
|  | IceBridge | 09.04.2015, 10.04.2015 | $1\,\mathrm{m} \times 1\,\mathrm{m}$ |
| SAR | ALOS-2 | 10.12.2017, 08.03.2023, 06.03.2024 | $4.3\,\mathrm{m} \times 3.4\,\mathrm{m}$ |

Further data was collected from various sources to support the study. The ENVEO velocity map provides data across Green-135 land with spatial resolution of $50\,\mathrm{m} \times 50\,\mathrm{m}$ (ENVEO, 2024), while PROMICE weather stations at the Russell glacier contributed local measurements of flow velocity, surface melt, surface and subsurface temperatures, and precipitation (Fausto et al., 2019).

## 3 SAR Methods

This section explains SAR polarimetry for identifying scattering processes, InSAR and TomoSAR to characterize and reconstruct vertical backscattering profiles.

### 3.1 SAR Polarimetry

Polarimetric SAR data enables qualitative and quantitative characterization of scattering mechanisms (Lee and Pottier, 2017). The systems considered operate in the horizontal (H) – vertical (V) transmit/receive basis, recording single-look complex





(SLC) scattering amplitudes $S_{HH}, S_{HV}, S_{VH}, S_{VV}$, where the subscripts indicate the transmit/receive configuration. In the case of a co-located transmitter and receiver and a reciprocal medium, it becomes $S_{HV} = S_{VH}$. To facilitate the analysis and

interpretation of scattering processes, the Pauli scattering vector $\mathbf{k}_P$ (Lee and Pottier, 2017) is typically used:

$$\mathbf{k}_P = \frac{1}{\sqrt{2}} \begin{bmatrix} S_{HH} + S_{VV} \\ S_{HH} - S_{VV} \\ 2S_{HV} \end{bmatrix}. \tag{4}$$

$S_{HH} + S_{VV}$ corresponds to the contribution of surface scattering (or more in general of odd bounces) to the total, $S_{HH} - S_{VV}$ to dihedral scattering (even bounces), and $S_{HV}$ to volume scattering. The intensities of these components are typically distributed across the channels of a single RGB image to provide a straightforward visual tool for a qualitative separation of different

scattering behaviors within a scene (Lee and Pottier, 2017).

For distributed scatterers, such as ice volumes, individual SAR resolution cells contain randomly distributed scattering contributions. Their behavior is better described using second-order statistics, with the $(3 \times 3)$ coherency matrix $\mathbf{T}$ derived from $\mathbf{k}_P$ with $\mathbf{T} = \langle \mathbf{k}_P \mathbf{k}_P^H \rangle$, where $\langle ... \rangle$ denotes spatial averaging over range-azimuth cells (i.e. multi-looking), and $(.)^H$ represents the Hermitian operator. A way to perform a quantitative polarimetric data analysis separates three orthogonal elementary scat-

tering mechanisms relying on the eigen-decomposition $\mathbf{T} = \mathbf{U}\mathbf{\Lambda}\mathbf{U}^{-1}$, in which $\mathbf{\Lambda}$ contains the eigenvalues $\lambda_1 \geq \lambda_2 \geq \lambda_3 \geq 0$ on the diagonal and zeros elsewhere, and $\mathbf{U} = [\mathbf{e}_1 \quad \mathbf{e}_2 \quad \mathbf{e}_3]$ contains three $3-$dimensional eigenvectors on the columns. The eigenvalues are used to calculate the entropy $0 \leq H \leq 1$, which measures scattering randomness or complexity of the scattering process (Cloude and Pottier, 1997):

$$H = -\sum_{i=1}^{3} p_i \log_3 p_i, \quad p_i = \frac{\lambda_i}{\sum_{i=1}^{3} \lambda_i}. \tag{5}$$

Low $H$ values indicate deterministic, single-scattering processes, whereas high values suggest multiple-scattering behavior. The polarimetric alpha angle (Cloude and Papathanassiou, 1998) can be calculated from the first element $e_{i1}$ of each eigenvector as $\alpha_i = \arccos(|e_{i1}|)$. Then, a statistical interpretation is obtained by calculating the mean alpha angle as $\alpha = \sum_{i=1}^{3} p_i \alpha_i$. Low values of $\alpha$ between $0°$ and $30°$ typically indicate surface scattering, intermediate values ($30° \leq \alpha \leq 60°$) volume scattering, and high values ($60° \leq \alpha \leq 90°$) dihedral scattering (Lee and Pottier, 2017).

## 3.2 SAR Interferometry

InSAR configurations use a limited number (typically two) of SLC SAR images $S_i(\omega)$ acquired under slightly different incidence angles (Fig. 2) to characterize the vertical backscattering profile within the same resolution cell. The three-dimensional (complex-valued) unitary vector $\omega$ represents the polarization state, defined as $S_i(\omega) = \omega^H \mathbf{k}_{Pi}$. For an interferometric pair ($i = 1, 2$), the complex InSAR coherence $\tilde{\gamma}_{\text{obs}}(\kappa_{zVol}, \omega)$ is the normalized cross-correlation of both images (Bamler and Hartl,

170 1998):

$$\tilde{\gamma}_{\text{obs}}(\kappa_{zVol}, \omega) = \frac{\langle S_1(\omega) S_2^*(\omega) \rangle}{\sqrt{\langle S_1(\omega) S_1^*(\omega) \rangle \langle S_2(\omega) S_2^*(\omega) \rangle}}. \tag{6}$$




The measured coherence can be decomposed into multiple factors accounting for various sources of decorrelation, including temporal, range spectral, and systematic effects ($\tilde{\gamma}_{\text{Tmp}}$, $\tilde{\gamma}_{\text{Sys}}$, $\tilde{\gamma}_{\text{Rg}}$). These factors depend on either $\kappa_{zVol}$ or $\omega$ (Zebker and Villasenor, 1992):

$$\tilde{\gamma}_{\text{obs}}(\kappa_{zVol}, \omega) = \tilde{\gamma}_{\text{Tmp}}(\omega) \cdot \tilde{\gamma}_{\text{Rg}}(\kappa_{zVol}) \cdot \tilde{\gamma}_{\text{Sys}}(\omega) \cdot \tilde{\gamma}_{\text{Vol}}(\kappa_{zVol}, \omega). \tag{7}$$

Finally, the volumetric decorrelation $\tilde{\gamma}_{\text{Vol}}(\kappa_{z\text{Vol}}, \omega)$ is related to the vertical backscattering profile $\sigma_V(z, \omega)$, where z indicates the height, by means of a Fourier relationship as (Bamler and Hartl, 1998):

$$\tilde{\gamma}_{\text{Vol}}(\kappa_{zVol}, \omega) = e^{i\kappa_z z_0} \frac{\int_{-\infty}^{0} \sigma_V(z, \omega) e^{i\kappa_{zVol} z} dz}{\int_{-\infty}^{0} \sigma_V(z, \omega) dz}, \tag{8}$$

in which $z_0$ indicates the topographic height of the glacier surface. $\sigma_V(z, \omega)$ is an electromagnetic quantity that depends on the vertical distribution (density) of surface and subsurface ice elements and their dielectric properties, and on the radar frequency, polarization and incidence angle (Bamler and Hartl, 1998). Moreover, the interferometric phase center height,

$$h_{pc}(\kappa_{zVol}, \omega) = \frac{\arg(\tilde{\gamma}_{\text{obs}}(\kappa_{zVol}, \omega))}{\kappa_{zVol}}, \tag{9}$$

where $\arg(.)$ is the argument of a complex number, approximates the centroid height of the vertical backscattering profile, especially for small $\kappa_{zVol}$ (Dall, 2007). In Sect. 4.1, phase center heights are computed relative to an external DEM, estimating the glacier surface topography, i.e. after compensating the related interferometric phase term from $\tilde{\gamma}_{\text{obs}}(\kappa_{zVol}, \omega)$. Moreover, estimating $\tilde{\gamma}_{\text{Vol}}(\kappa_{zVol}, \omega)$ is crucial for characterizing sub-surface scattering. For the airborne interferometric acquisitions (Table 2), non-volumetric decorrelation contributions are negligible due to the short acquisition time, large bandwidth, low noise, and high interferometric processing accuracy. Therefore, the complex InSAR coherence is assumed to be solely influenced by volumetric effects (Eq. 8), without any additional compensation for other sources of decorrelation.

## 3.3 SAR Tomography

TomoSAR techniques reconstruct the vertical backscattering profile by combining SAR images from multiple incidence angles (Reigber and Moreira, 2000). The TomoSAR data vector $\mathbf{y}(\omega) = [S_1(\omega), S_2(\omega), \ldots, S_K(\omega)]^T$ is constructed, where $K$ is the number of SAR images (Fig. 2), e.g., $K = 7, 8$ for the F-SAR data sets at L- and P-band (Table 2). The TomoSAR data vector is then used to compute the $(K \times K)$ TomoSAR covariance matrix $\mathbf{R}(\omega)$ as:

$$\mathbf{R}(\omega) = \langle \mathbf{y}(\omega) \mathbf{y}^H(\omega) \rangle. \tag{10}$$

Assuming negligible non-volumetric decorrelation, each element of $\mathbf{R}(\omega)$ is related to $\sigma_V(z, \omega)$ through a Fourier relationship with the vertical wavenumber (Eq. 2). In absence of scattering models, $\sigma_V(z, \omega)$ is reconstructed as (Lombardini and Reigber, 2003; Stoica and Moses, 2005):

$$F(z, \omega) = \mathbf{h}^H(z, \omega) \mathbf{R}(\omega) \mathbf{h}(z, \omega), \tag{11}$$





where $\mathbf{h}^H(z,\omega)$ is a (polarization- and height-dependent) coefficient vector preserving backscattering at height $z$ while atten-
uating other contributions. The analysis in Sect. 4.3.1 follows the Capon method (Lombardini and Reigber, 2003; Stoica and
Moses, 2005):

$$\mathbf{h}(z,\omega) = \frac{\mathbf{R}^{-1}(\omega)\mathbf{a}(z)}{\mathbf{a}^H(z)\mathbf{R}^{-1}(\omega)\mathbf{a}(z)}, \tag{12}$$

where the $K$-dimensional steering vector $\mathbf{a}(z) = \left[1, e^{-j\kappa_{zVol,2}z}, \ldots, e^{-j\kappa_{zVol,K}z}\right]^T$ is a function of the interferometric phase
generated by a scattering contribution at height $z$ across all the acquisitions with respect to a reference one.

Here, $\kappa_{zVol,k}$ is the vertical wavenumber for the $k$-th track with respect to a reference one. The Capon formulation adapts
for each height to the data through $\mathbf{R}(\omega)$, hence it leads to reconstructions of $\sigma_V(z,\omega)$ with vertical resolution better than the
limit $\rho_z$ in Eq. 3 imposed by the track geometry. The inherent loss of radiometric accuracy (Lombardini and Reigber, 2003;
Cazcarra-Bes et al., 2020) induced by the increase of resolution is deemed of secondary importance for the analysis in this
paper.

The TomoSAR data vector can also be processed in order to isolate scattering contributions within a height interval and
reconstruct their vertical backscattering profile as detailed above. This filtering is here performed using a $(K \times K)$-dimensional
(complex-valued) matrix $\mathbf{G}$ (Lombardini and Pardini, 2009; Joerg et al., 2017):

$$\mathbf{y}_f(\omega) = \mathbf{G}\mathbf{y}(\omega). \tag{13}$$

The derivation of the matrix filter $\mathbf{G}$ requires first of all the definition of a height interval in which backscattering contributions
are left undistorted, and a height interval in which backscattering contributions are cancelled. Then, the elements of $\mathbf{G}$ are
calculated from a least-squares optimization involving a set of steering vectors calculated for sampled heights in the undistorted
and attenuated intervals. The complete derivation and formulation is reported in Joerg et al. (2017).

## 4   Characterization of Ice Features

This section investigates the radar-dark and radar-bright features in our SAR data, examining their scattering processes, sur-
face and subsurface characteristics, and temporal stability to understand their origins and potential glaciological formation
processes.

### 4.1   Analysis of Polarimetric and Interferometric Data

We begin by examining scattering mechanisms in radar-dark and radar-bright features, using Sentinel-2 optical imagery and
multi-frequency Pauli representations (X-, L-, and P-band) for the Russell Glacier test site. The Pauli (Eq. 4) representations,
in range-azimuth coordinates, offer qualitative insights into scattering mechanisms, visualizing surface scattering (HH+VV)
in blue, dihedral scattering (HH-VV) in red, and volume scattering (HV) in green. By combining optical and SAR data, we
integrate visual and scattering information to enhance our understanding of the site's characteristics. The test site was processed
in two parts for X- and L-band and color-scaled separately, then recombined for analysis.





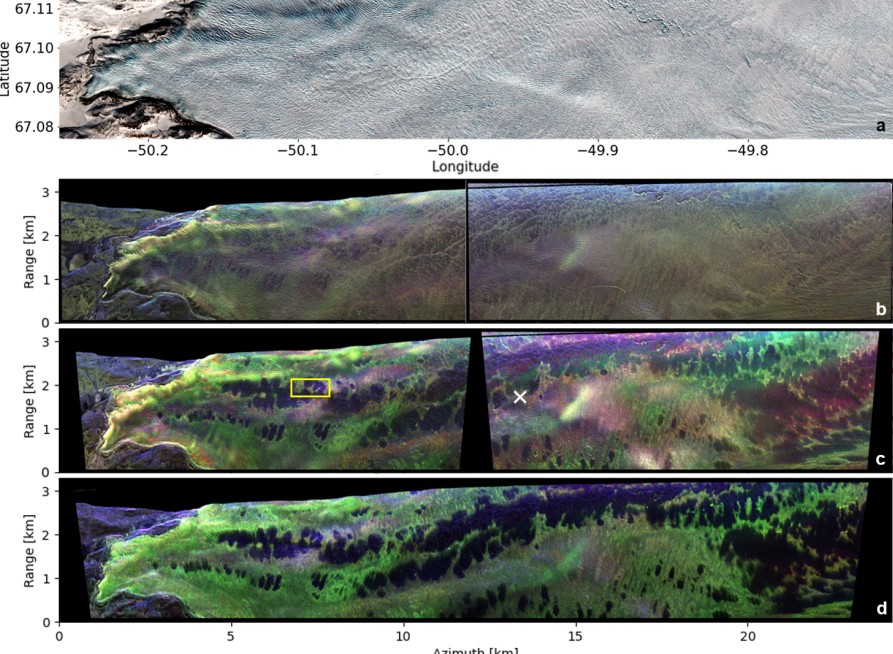

**Figure 3.** (a) Optical data (Sentinel-2) in geo-coordinates and polarimetric data in range-azimuth acquired in (b) X-, (c) L- band with a yellow rectangle indicating the area for surface characterization discussed in Sect. 4.2, and a white cross marks the location of the KAN L weather station, as detailed in Sect. 4.3.2., (d) P-band at the Russell glacier and shown in the Pauli basis (HH+VV (surface scattering) in blue, HH-VV (dihedral scattering) in red, HV (volume scattering) in green).

Fig. 3 shows the extent of the test site, with the glacier terminus at the western end, surrounded by rocky terrain. The minimal snow cover on the glacier reveals the overall topography and surface roughness, including small and large crevasses in the optical image in Fig. 3a. In the X-band Pauli image in Fig. 3b, these crevasses are represented as green lines. Their appearance may shift to red, indicating dihedral scattering, depending on their orientation relative to the radar's flight direction. Bright green areas usually signify volume scattering, associated with high surface roughness, highly crevassed areas, or potentially shallow penetration into the ice. L-band data (Fig. 3c) penetrates deeper into the ice (Table 5), providing more subsurface information while reducing surface detail. Here, radar-dark features dominated by surface scattering sharply contrast with radar-bright features characterized by volume scattering. In P-band data (Fig. 3d), the scattering patterns are similar to L-band, with enhanced subsurface contrast, though at a lower resolution (Table 2). By comparing HV backscatter in Fig. 1 with the Pauli representation in Fig. 3c, radar-dark features are linked to low backscatter and surface scattering, while radar-bright features correspond to high backscatter and volume scattering. However, not all volume scattering areas correspond strictly to radar-bright features, as highly crevassed areas may also appear as stronger volume scattering than surrounding regions.

Since Pauli representations are qualitative, we further analyze scattering entropy $H$ and mean alpha angle $\alpha$ to quantify scattering mechanisms for radar-dark and radar-bright features across X-, L-, and P-band (Eq. 5), where both features exhibit





distinct characteristics with dominant scattering mechanisms (Table 4). Radar-dark features have low entropy values in L-
and P-band (0.0 to 0.5) and low mean alpha angles (0° to 25°), indicating surface-dominated scattering with minimal volume
contribution. X-band, however, shows a broader range for $H$ and $\alpha$, likely due to different interactions with surface roughness.
In contrast, radar-bright features have higher entropy (up to 1.0) and mean alpha angles (up to 60°) across all frequency bands,
indicating multiple scattering mechanisms, including volume scattering, that are typical of more heterogeneous or layered
subsurface structures (Parrella et al., 2021).

**Table 4.** Entropy $H$ and mean alpha angle $\alpha$ range for radar-dark and radar-bright features across X-, L-, and P-band.

| Parameter | Frequency band | Radar-dark features | Radar-bright features |
|---|---|---|---|
| | X-band | 0.2 to 0.8 | 0.2 to 0.9 |
| Entropy ($H$) | L-band | 0.2 to 0.5 | 0.5 to 1.0 |
| | P-band | 0.0 to 0.4 | 0.6 to 1.0 |
| | X-band | 0° to 40° | 0° to 60° |
| Mean alpha angle ($\alpha$) | L-band | 0° to 20° | 20° to 45° |
| | P-band | 0° to 25° | 30° to 60° |

After using PolSAR to identify scattering mechanisms, we shift our focus to InSAR, which complements this analysis by
providing phase center heights (Eq. 9) from two acquisitions with a $10\,\mathrm{m}$ horizontal displacement between their tracks, allowing
us to gain vertical information that PolSAR cannot capture. A change over different horizontal displacements and therefore a
change in $\kappa_{zVol}$ values (Eq. 2) will be shown in Sect. 4.4 for InSAR coherences.

**Table 5.** Phase center heights for radar-dark and radar-bright features across X-, L-, and P-band.

| Frequency band | Polarization | Radar-dark features (height range) | Radar-bright features (height range) |
|---|---|---|---|
| X-band | HH | $-2\,\mathrm{m}$ to $0\,\mathrm{m}$ | $-2\,\mathrm{m}$ to $0\,\mathrm{m}$ |
| | HV | $-3\,\mathrm{m}$ to $0\,\mathrm{m}$ | $-5\,\mathrm{m}$ to $0\,\mathrm{m}$ |
| L-band | HH | $-2\,\mathrm{m}$ to $0\,\mathrm{m}$ | $-15\,\mathrm{m}$ to $-10\,\mathrm{m}$ |
| | HV | $-5\,\mathrm{m}$ to $0\,\mathrm{m}$ | $-30\,\mathrm{m}$ to $-20\,\mathrm{m}$ |
| P-band | HH | $-4\,\mathrm{m}$ to $0\,\mathrm{m}$ | $-35\,\mathrm{m}$ to $-20\,\mathrm{m}$ |
| | HV | $-6\,\mathrm{m}$ to $0\,\mathrm{m}$ | $-50\,\mathrm{m}$ to $-30\,\mathrm{m}$ |

Table 5 demonstrates that radar-dark features consistently exhibit shallow signal penetration depths, with negative phase
center heights across all frequency bands and polarizations, typically ranging between $-6\,\mathrm{m}$ to $0\,\mathrm{m}$. This limited depth aligns
with PolSAR analysis, which indicates dominant surface scattering with minimal contributions from subsurface structures or
volume scattering. In contrast, in L- and P-band, radar-bright features reach significantly lower phase center heights, extending
from $-50\,\mathrm{m}$ to $-15\,\mathrm{m}$. This significant difference suggests that something is inhibiting further penetration into the ice for




radar-dark features. It may be due to increased attenuation caused by changes in permittivity, likely from water content or

impurities in the ice, or smooth surfaces reflecting the radar signal, limiting penetration through strong surface reflectance
(Dall et al., 2001), which will be explored further in Sect. 4.2 and 4.3.

## 4.2    Surface Characterization

### 4.2.1    Analysis of Topography and Surface Roughness

After reviewing PolSAR and InSAR scattering processes, we analyzed whether radar-dark features correlate with surface

processes by examining glacier topography and roughness. Surface roughness, derived from a high-resolution orthophoto (Fig.
4a) and ATM lidar data from NASA IceBridge (Studinger et al., 2022), highlights surface variability in a central sample area
(yellow rectangle in Fig. 3c). HV L-band backscatter (Fig. 4b), processed with a $10\,\mathrm{m} \times 10\,\mathrm{m}$ multi-looking window, was
used to locate these features. The temporal proximity of L-band and IceBridge data (Tables 1 and 3) ensures minimal surface
changes, allowing for better comparability. Additionally, the reference DEM, acquired during the same F-SAR campaign,

was analyzed to assess surface topography across the sample area. Given the overall slope in the sample area, the DEM was
de-trended using a $50 \times 50$ window smoothing filter, and the difference from the original DEM was computed to emphasize
small-scale topographic variations (Fig.4c).

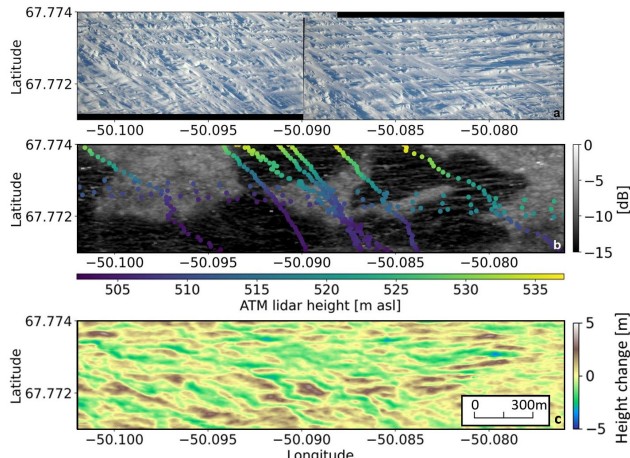

**Figure 4.** Surface characterization of the sample area: (a) orthophoto by NASA IceBridge, (b) L-band, HV backscatter (grey-scale) with
overlaid point-wise elevation of ATM lidar data (color-scale) from NASA IceBridge, (c) de-trended reference DEM derived from DLR's
F-SAR data

The orthophoto (Fig. 4a) reveals a snow-covered glacier surface characterized by grid-like surface roughness patterns, which
may result from intersecting strain fields or meltwater channels (Hambrey and Lawson, 2000; Yang et al., 2021), or alterna-

tively, may represent wind-formed features. The ATM lidar tracks overlaid on the HV L-band backscatter image (Fig. 4b) show
elevation differences along the flight lines ($500\,\mathrm{m}$ to $540\,\mathrm{m}$ elevation), illustrating slope changes and overall surface variability



within the sample area. While the lidar tracks suggest subtle elevation gradients across the area, the de-trended DEM (Fig. 4c) emphasizes small-scale topographic variations, revealing a pattern broadly resembling the surface roughness observed in the orthophoto. However, despite these surface features, no direct spatial correlation is evident between the grid-like roughness patterns and radar-dark or radar-bright features, presented in the HV L-band data (Fig. 4b), suggesting that the observed radar signature is not primarily controlled by surface roughness or topography (also at a larger scale, no correlation was found over the complete test site).

### 4.2.2 Analysis of Albedo Dynamics

Building upon the previous analysis, examining albedo dynamics provides deeper insight into surface composition. Higher albedo values indicate cleaner, brighter surfaces, such as fresh snow, which reflect more sunlight, whereas lower values correspond to darker, impurity-rich surfaces that absorb more heat (Navari et al., 2021). During the campaign acquisition period, albedo values measured at the KAN L Automatic Weather Station ranged from 0.7 to 0.9, indicating the presence of a snow cover (Fausto et al., 2019), consistent with the optical orthophoto shown in Fig. 4a.

Tedstone and Cook (2020) investigated albedo in relation to surface types and utilized Sentinel-2 data from summer 2017 to generate a comprehensive albedo map of the Russell Glacier, exhibiting an overall albedo range of 0.1 to 0.7. Radar-dark features emerge with their distinct shapes (Fig. 1) as areas of slightly lower albedo compared to the surrounding radar-bright features within the sample area, indicated by a red rectangle (Fig. 5). However, closer to the terminus, distinguishing albedo variations unrelated to topography and surface roughness becomes increasingly challenging.

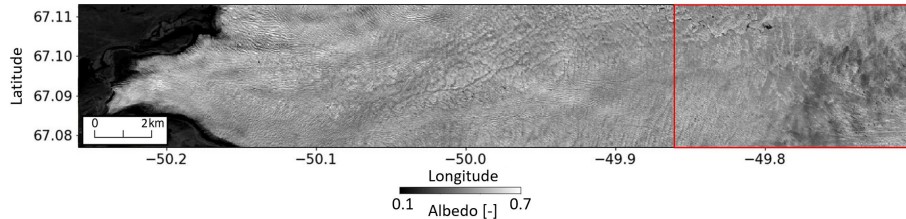

**Figure 5.** Albedo map modified after Tedstone and Cook (2020), derived from Sentinel-2 imagery acquired on July 21, 2017. Low-albedo areas indicate cloud coverage, while the shape of radar-dark features is distinctly visible within the red rectangle as regions with lower albedo.

Furthermore, Ryan et al. (2018) classified Russell Glacier surface types into distinct categories during summer 2015, identifying two key types pertinent to this study: clean ice and ice containing uniformly distributed impurities. Although their point-based measurements do not encompass the entire surface, their findings reinforce the hypothesis that radar-bright features may correspond to clean ice, while radar-dark features are likely associated with impurity-rich areas.

Impurity-rich areas have also been associated with the formation and persistence of a weathering crust, as demonstrated by Tedstone et al. (2020), which plays a critical role in enhancing meltwater retention and altering ice surface properties, contributing to the weathering crust's development. The presence of residual liquid water within this crust, even during winter, may lead to near-surface attenuation of the radar signal, thereby explaining the observed SAR characteristics. This relationship





suggests that radar-dark features may indicate areas of crust development, while radar-bright features likely represent cleaner ice, free of residual liquid water, which permits deeper radar penetration (Sect. 4.1). If this hypothesis holds, then SAR analysis significantly enhances the visibility of areas with weathering crust.

### 305    **4.3   Subsurface Characterization**

#### 4.3.1   Analysis of Tomographic and Sounder Data

The tomographic analysis in P-band provides valuable insights into subsurface scattering at the Russell glacier. The RGB Pauli representation (Fig. 6a) clearly illustrates the transitions between radar-bright and radar-dark features along the white transect line, where Sounder and tomographic profiles are generated. Fig. 6b shows HV Sounder data in P-band. Polarization

variability in TomoSAR is illustrated by comparing HH and HV, as shown in Fig. 6c-d using Capon tomograms (Eqs. 11 and 12), with VV omitted due to its similarity with HH. In Fig. 6e, another HV Capon tomogram is shown, generated using a matrix filter (Eq. 13), which removes scattering within a height interval of $-4\,\mathrm{m}$ to $2\,\mathrm{m}$ relative to the reference DEM, enhancing subsurface scattering visibility (Sect. 3.3). All Capon tomograms in Fig. 6c-e are normalized for each tomographic profile along azimuth, ranging from minimum to maximum values (0 to 1), offering a qualitative representation of the vertical

backscattering distribution across the transect line.

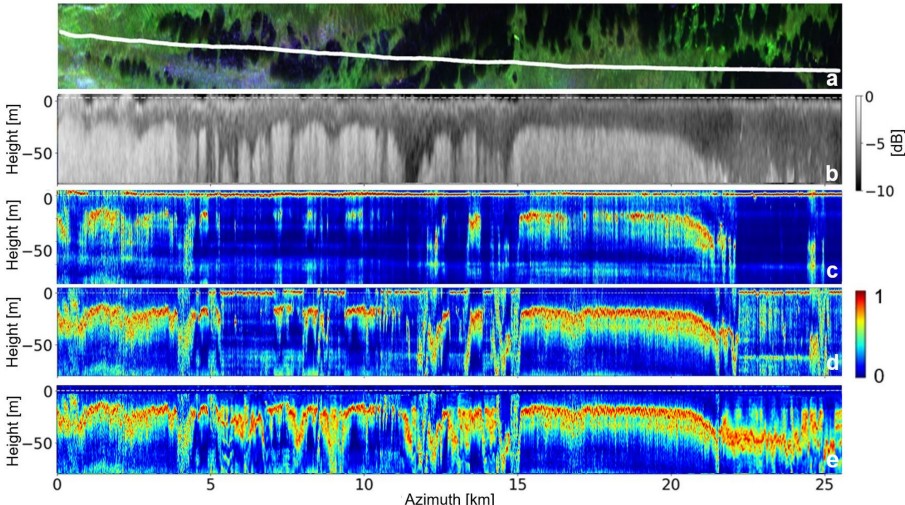

**Figure 6.** (a) RGB Pauli representation with the white transect line for Sounder and tomographic profiles, (b) Sounder data (nadir-looking), (c) Capon tomogram (side-looking) in HH, (d) Capon tomogram (side-looking) in HV, and (e) Capon tomogram (side-looking) in HV with surface scattering components removed. All plots are shown in P-band. The height for c-e is set with respect to the surface elevation from the reference DEM.

In the Capon tomograms, a clear separation between the two ice features can be observed, especially in HV (Fig. 6d). Here, radar-bright features exhibit a well-defined, continuous subsurface scattering layer at heights between $-15\,\mathrm{m}$ to $-50\,\mathrm{m}$, which



remains consistent and prominent across the transect line, indicating a robust subsurface structure. Tomographic analyses at various locations confirmed the presence of this consistent subsurface scattering layer across the entire test site. The HH tomogram (Fig. 6c) reveals a similar subsurface scattering layer, though with more pronounced surface scattering that reduces the visibility of the subsurface layer. For radar-dark features, the HV tomogram shows primarily surface scattering with minimal subsurface scattering. The HH tomogram further emphasizes the scattering at the surface, further diminishing the visibility of any possible subsurface scattering structures. Due to the poor visibility of these potential subsurface scattering structures for radar-dark features in both HH and HV polarizations, we applied the matrix filter to enhance subsurface scattering in Fig. 6e. This removal of surface scattering primarily affects the radar-dark features, as surface scattering components in HV for radar-bright features are already minimal. Here, the weak subsurface scattering, now revealed for radar-dark features, suggests the presence of a subsurface layer similar to the radar-bright features, although it remains less pronounced.

To further understand these subsurface scattering structures for both ice features, the Sounder data (Fig. 6b) serves as a valuable comparison. For radar-bright features, the Sounder HV data aligns well with the findings in Fig. 6d, revealing a consistent strong radar return starting at heights between $-15\,\mathrm{m}$ to $-50\,\mathrm{m}$, closely matching the main subsurface scattering layer observed in the tomograms. The downwards extent of this layer in the Sounder data could be due to off-nadir returns of the same, rather compact, layer as in the tomograms, or due to an actual vertical extent of the strong radar returns. For radar-dark features, the Sounder data reveals much deeper locations of the strong radar return, down to $-100\,\mathrm{m}$, which occurs often deeper than what is detectable in the matrix-filtered tomogram in Fig. 6e. This suggests that radar-dark features may indeed contain deeper subsurface structures that are weak but still observable, though the evidence is less conclusive compared to the radar-bright features.

Overall, the tomographic and Sounder analysis reveals distinct subsurface characteristics between radar-bright and radar-dark features and confirm the PolSAR and InSAR understanding of dominant surface scattering at the radar-dark features and subsurface volume scattering at the radar-bright features. The presence of weak but observable subsurface scattering in the radar-dark features indicates that englacial water bodies, such as water pockets or channels are unlikely to be the cause of these features. If significant liquid water were present, the radar signal would be fully absorbed, preventing any subsurface scattering response. However, a small amount of liquid water within the ice could still explain an attenuation of the signal, causing the low-backscatter areas observed in Fig. 1 for the radar-dark features. In contrast, the well-pronounced subsurface scattering layer within the radar-bright features suggests these areas are frozen, with no water inclusions or remaining liquid water content, at least down to the depth of the main subsurface scattering layer.

### 4.3.2 Analysis of Temperature Profiles

Subsurface temperature profiles recorded by the KAN L Automatic Weather Station (Fausto et al., 2019) at the Russell glacier offer valuable insights into seasonal and vertical temperature variations within the radar-bright feature area. Nine temperature measurements were taken at depths ranging from $-10\,\mathrm{m}$ to $0\,\mathrm{m}$, relative to the ice surface, as illustrated in Fig. 7. The exact depths may vary due to accumulation and ablation processes (Fausto et al., 2019).




Fig. 7 highlights notable temperature variations at all available measurement depths during 2015. Below, subsurface temperatures are expected to be more consistent, suggesting a stabilization of thermal conditions at deeper levels. However, the lack of deeper measurements between $-50\,\mathrm{m}$ to $-15\,\mathrm{m}$ presents a limitation for comparison, as this range corresponds to the main subsurface scattering layer identified in SAR tomography and Sounder data (Sect. 4.3.1). Despite this constraint, it is reasonable to assume that the ice in the upper layers undergoes significant seasonal temperature fluctuations, whereas the conditions below remain comparatively stable. This transition zone may represent a thermal boundary that is detectable in both tomography and Sounder data, manifesting as the main subsurface scattering layer.

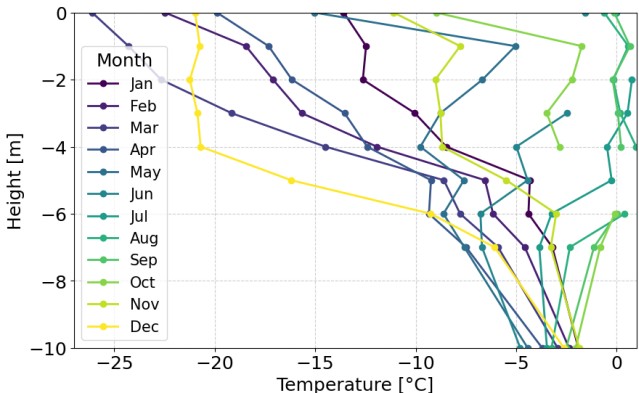

**Figure 7.** Monthly subsurface temperature profiles for the year 2015 by the KAN L Automatic Weather Station (Fausto et al., 2019). Its location within the test site is indicated in Fig. 3.

## 4.4 InSAR Modelling of Scattering Structures

Building on the surface and subsurface characterization of radar-dark and radar-bright features, this modeling approach aims to transfer the current understanding of the scattering structures into a quantitative InSAR structure model at example locations. For this purpose, established InSAR structure model components are combined to simulate coherence magnitudes and comparing them with those measured from different track combinations in both L- and P-band. This exploits the sensitivity of InSAR coherences to volume decorrelation due to the vertical distribution of scattering (Eq. 8), as explained in Sect. 3.2. A model-based analysis of the entire test site goes beyond the scope of this paper, therefore, an exemplary location, defined as a single, multi-looked pixel, was selected for both radar-dark and radar-bright features. Detailed formulas and modeling parameters used in this analysis are provided in Appendix A.

For the modeling of the radar-bright features, a two-component model is applied, one for the surface and one for the subsurface, due to the clear separation of scatterers in the tomographic analysis (Sect. 4.3.1). Hereby, the surface component is modeled using a rectangular function with a height of $2\,\mathrm{m}$ around the ice surface, which is based on the reference DEM, to account for the surface roughness on the glacier (Sect. 4.2). The subsurface component is modeled using a Uniform Volume (UV) model, which assumes a uniform distribution of scatterers within the volume (Zebker and Hoen, 2000; Fischer et al.,




2020), starting at a height of $-15\,\mathrm{m}$ for L- and P-band. The coherence magnitude of this two-component model is simulated across different $\kappa_{zVol}$ values, representing various track displacements, and incorporates a surface-to-volume ratio $m(\omega)$ that changes with polarization.

Fig. 8a-b present the results of modeling radar-bright features in L-band, while Fig. 8c-d show the corresponding results for P-band. In both cases, the simulated coherence is represented by a line, and the measured coherences by discrete points. The general drop of coherence with increasing track displacement and $\kappa_{zVol}$ shows the expected volume decorrelation, whereas the undulating behavior is typical for two scattering components with a certain vertical distance. The two-component model generally provides a good fit across both frequency bands and polarizations.

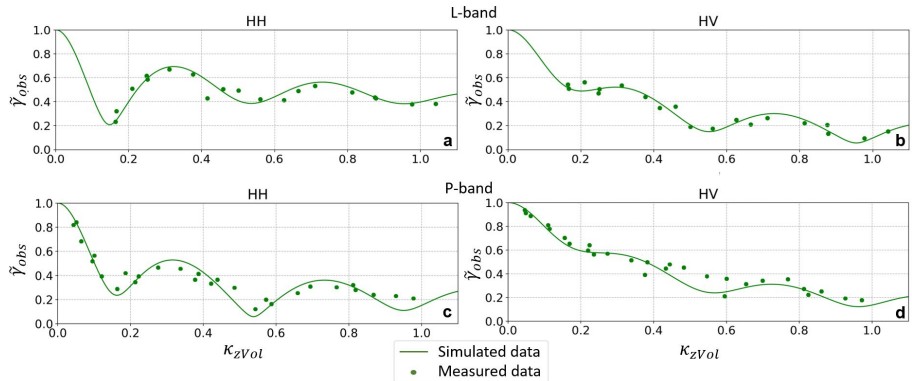

**Figure 8.** Modeling of coherence magnitudes at the representative location of radar-bright features in different polarizations (HH and HV) for L-band (a-b) and P-band (c-d). The points represent the measured coherence, while the lines represent the simulated coherence.

Modeling the radar-dark features presents additional challenges due to their less distinct subsurface scattering structures. Previous analyses suggest that radar-dark features are primarily dominated by surface scattering, with minimal subsurface contributions. These subsurface contributions could be enhanced in Fig. 6e, which reveals a weak subsurface scattering layer at similar heights to those observed for radar-bright features. However, the Sounder data in Fig. 6b suggests a lower height for the subsurface scattering layer than in the radar-bright features. In this case, a three-component model can match the surface

and subsurface scattering processes observed in both frequency bands and polarizations. The surface component is modeled identically as for the radar-bright features, supporting that surface properties are independent of the formation of radar-dark and radar-bright features. The main subsurface scattering layer, detected in Fig. 6e, is also modeled using a UV model with adjusted parameters (see Appendix A). Additionally, to fit the measured coherences, we added a weak subsurface layer at $-5\,\mathrm{m}$, even though this is generally not indicated by the tomograms.

Overall, Fig. 9 demonstrates a good match between the modeled and measured data for both frequency bands and polarizations, providing insights into the scattering processes of radar-dark features. In the L-band HH polarization (Fig. 9a), coherence values remain consistently high across different $\kappa_{zVol}$ values, gradually decreasing from 1 to 0.8 as $\kappa_{zVol}$ increases, as expected for dominant surface scattering. Still, the two subsurface components are also required to match this particular coherence




decrease. In contrast, the P-band HH polarization (Fig. 9c) exhibits a steeper decline in coherence with increasing $\kappa_{zVol}$ values,

indicating higher volume decorrelation due to more penetration into the ice. For HV polarization in both frequency bands (Fig. 9c-d), the coherence shows significant variability over changing $\kappa_{zVol}$ values, suggesting a complex subsurface structure. We purposely focused the simulation on capturing the higher-order oscillations of the coherences in the HV polarization to gain an understanding of the required complexity in the vertical scattering structure.

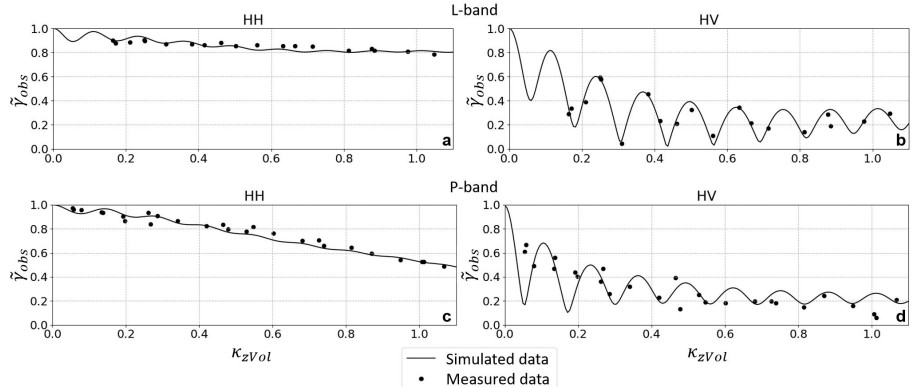

**Figure 9.** Modeling of coherence magnitudes at the representative location of dark features in different polarizations (HH and HV) for L-band (a-b) and P-band (c-d). The points represent the measured coherence, while the lines represent the simulated coherence.

In summary, modeling of radar-bright features confirmed the surface and subsurface scattering structures identified in previ-

ous analyses. Moreover, we confirmed that the surface characteristics are independent of the presence of radar-bright or radar-dark features. However, for the radar-dark features, the subsurface scattering structure remains inconclusive, with tomographic, Sounder, and modeling analyses each suggesting different heights for potential subsurface scattering layers. Nonetheless, the UV model, along with an additional weak subsurface layer, matched the measured coherences very well as a two-component, subsurface scattering structure for the representative location of the radar-dark features.

### 405   4.5   Temporal Stability of Ice Features

Building on the surface and subsurface characterization of ice features, we introduce a temporal component to further investigate their formation processes or origins. Hereby, we focus on the temporal changes of radar-dark features within our test site at the Russell glacier, using time series data from ALOS-2 shown in Table 3 (JAXA, 2024). Fig.10 presents backscatter images in HV, showing the test site under frozen conditions in 2017, 2023, and 2024 (Fig. 10a-c). A backscatter threshold of $-10\,\mathrm{dB}$

is used to outline the radar-dark features, which are overlaid in Fig. 10d for 2023 and 2024. In Fig. 10d-f, we highlight three sample areas that demonstrate variations in these features from 2017 to 2024, offering insights into potential changes over time.

The overall description of the radar-bright and radar-dark features in the HV backscatter images (Fig. 10a-c) is similar to Fig. 1. The radar-dark features are visible as low-backscatter areas on the glacier. The radar-bright features, representing high-backscatter areas, surround the radar-dark features on the glacier. A visual comparison of the three HV backscatter images





(Fig. 10a-c) suggest that radar-dark features remain stable in shape, size and position over the observed period and almost no change can be detected. By overlaying the outlines of the radar-dark features from 2023 to 2024, some changes, although subtle, are visible. Comparing the velocity maps from ENVEO (2024) with Fig. 10d confirms that the glacier moves slowly, with an average velocity of about $100 \, \mathrm{m\,y^{-1}}$ at the eastern side of the test site (van de Wal et al., 2015), slowing down towards the terminus. Hereby, the radar-dark features generally move at the same velocity and in the same direction as the glacier.

Analyzing the sample areas (Fig. 10e-g) shows the different changes between 2017 and 2024.

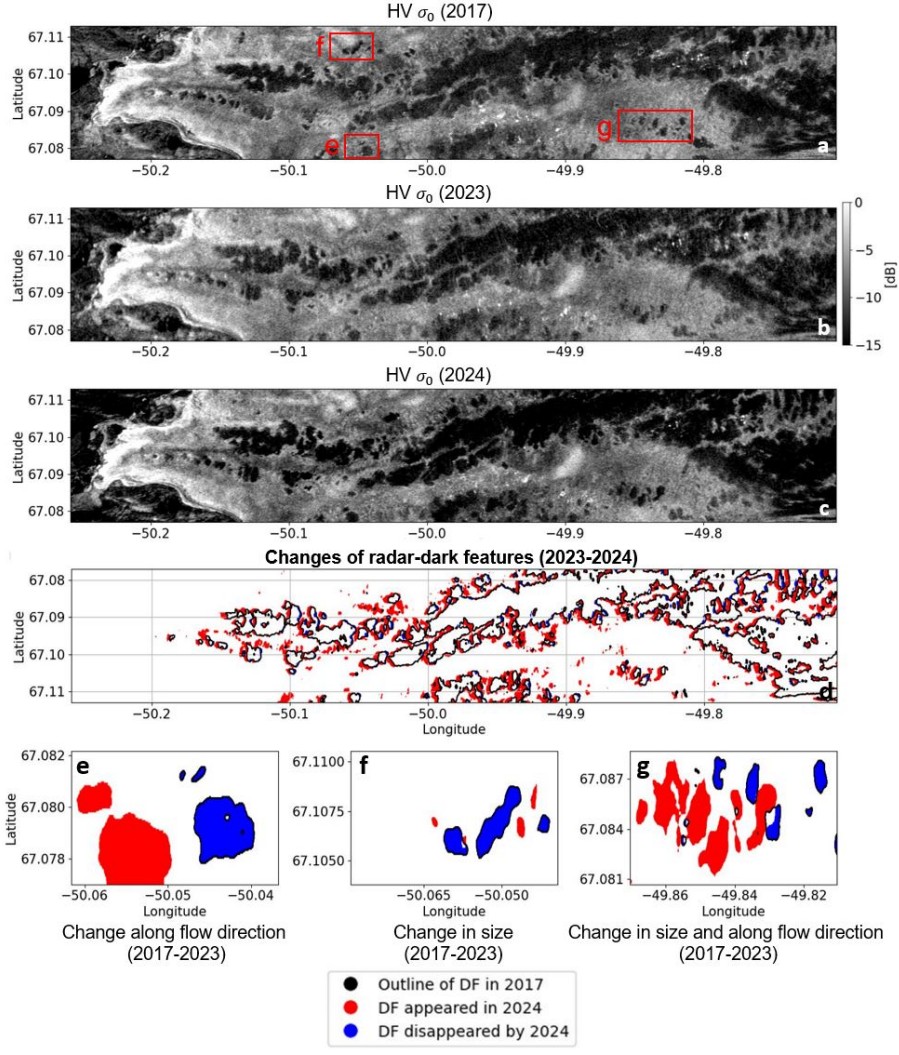

**Figure 10.** ALOS-2 HV backscatter data from (a) 2017, (b) 2023 and (c) 2024, (d) the change and movement of the radar-dark features between 2023 in blue and 2024 in red; (e-f) three sample areas to display the variations of change between 2017 in blue and 2024 in red.



In Fig. 10e, a radar-dark feature observed in 2017 (shown in blue) has moved consistent with the glacier's velocity by 2024, with a slight increase in size. Additionally, two smaller features in 2017 have merged, forming a larger composite feature by 2024 (shown in red). In Fig. 10f, three distinct radar-dark features are visible in 2017. By 2024, the two larger features have nearly disappeared, while the smallest feature remains stable and continues to move with the glacier. In Fig. 10g, radar-dark features have increased in size and moved with the glacier over the observed period, with new radar-dark features also forming in this time period. These results show that radar-dark features move with the glacier's velocity and direction, showing generally a high temporal stability, but also varying behaviors over time. These behaviors include no change, as well as significant increases or decreases in their size. Given the significant surface glacier melt (van de Wal et al., 2015; ENVEO, 2023), as described in Sect. 2, their consistent shape and persistence raise questions about their origin and formation processes.

## 5 Discussion

### 5.1 Radar-Dark Features

Radar-dark features exhibit dominant surface scattering across multiple frequencies (X-, L-, and P-band) (Sect. 4.1), with no link to topography or surface roughness, suggesting a subsurface origin (Sect. 4.2). They generally have an oval or round shape, with a mean size of several hundred meters, and often form interconnected patterns (Fig. 1). InSAR observables (Sect. 4.1) and SAR tomography (Sect. 4.3.1) confirm scattering mainly at the surface with weak contributions from the subsurface, which also could be modeled in Sect. 4.4. Moreover, analysis of ALOS-2 data revealed that the radar-dark features move with the glacier flow and remain relatively stable in shape, with only minor changes over time (Sect. 4.5), despite the large surface melt. The combined SAR analyses suggest that deeper penetration in L- and P-band is obstructed, likely due to high signal attenuation just below the surface. The attenuation is probably caused by small amounts of liquid water within the ice. However, significant water inside the glacier, such as englacial water bodies, would fully absorb the signal, eliminating any subsurface scattering contributions.

### 5.2 Radar-Bright Features

Radar-bright features are characterized by high backscatter in both L- and P-band, with volume scattering as the dominant scattering mechanism (Sect. 4.1). InSAR analysis (Sect. 4.1) and SAR tomography confirm deeper penetration and a prominent subsurface scattering layer between $-15\,\mathrm{m}$ to $-50\,\mathrm{m}$, extending across the entire test site. Moreover, Sounder data confirmed this subsurface scattering layer for the radar-bright features (Sect. 4.3.1). Modeling results further suggest that weak surface contributions exist, but they are less significant compared to the dominant subsurface scattering in the radar-bright features (Sect. 4.4), especially in HV polarization.





### 5.3 Preliminary Glaciological Interpretation

**5.3.1 Comparison of Radar-Dark and Radar-Bright Features**

Radar-dark and radar-bright features share several key characteristics, including a lack of correlation with glacier topography or surface roughness (Sect. 4.2.1). Additionally, both ice features are similarly affected by the high surface melt rate of approximately $2.5 \, \mathrm{m \, y^{-1}}$ to $5.6 \, \mathrm{m \, y^{-1}}$ (van de Wal et al., 2012; ENVEO, 2023). Investigated glaciological phenomena such as drained or refrozen supraglacial lakes (Hu et al., 2021; Yang et al., 2021), the dark zone—an area located further inside the

Greenland ice sheet—and cryoconites, which may exist but are limited to very small scales and do not reach the size of the radar-dark features (Wientjes and Oerlemans, 2010; Ryan et al., 2018), as well as englacial water bodies (Catania et al., 2008; Lampkin and VanderBerg, 2013), have been ruled out as potential causes.

The most plausible explanation remaining is that the radar-dark features are influenced by a weathering crust (Cooper et al., 2018). The weathering crust, a porous layer formed near the glacier surface through meltwater infiltration and refreezing

processes, plays a crucial role in shaping the near-surface properties of the ice (Cooper et al., 2018) and causes lower albedo values in summer due to a higher fraction of impurities (Sect. 4.2.2). This process may explain the formation and behavior of radar-dark features, contributing to their distinctive characteristics. During winter, the weathering crust can trap residual liquid water in the ice just below the surface, leading to signal attenuation, which would reduce the subsurface scattering contribution and overall backscatter in the radar-dark features significantly.

While the weathering crust theory aligns with most observations, it does not explain the shape and long-term stability of radar-dark features (Sect. 4.5). A possible explanation for the stable shape could be a connection to past supraglacial lake extent, as the radar-dark features strongly resemble lake outlines. The increased deposit of sediments at the location of past supraglacial lakes (Selmes et al., 2013; Leidman et al., 2023) might enhance the formation process of the weathering crust (Cooper et al., 2018). In this sense, past supraglacial lakes could drive the spatial pattern of the weathering crust. As lakes tend

to form in consistent locations, their residuals, shaped by glacier flow, could produce the spatial patterns observed in our data. However, no direct evidence was found to support this theory.

In contrast, radar-bright features could correspond to bare, clean ice without the presence of the weathering crust. In these areas, there is minimal or no meltwater infiltration into the ice subsurface, resulting in mostly surface runoff and solid refreezing in winter, with no residual liquid water content (Cooper and Smith, 2019). This allows for high penetration of the SAR signal

into the ice, particularly at lower frequencies, as shown in the interferometric (Sect. 4.1) and tomographic analysis (Sect. 4.3.1) of the radar-bright features. The enhanced penetration results in strong (volume) backscatter from any structural or dielectric heterogeneities in the ice subsurface of the radar-bright features.

**5.3.2 Interpretation of the Main Subsurface Scattering Layer**

The subsurface scattering layer in the radar-bright features is prominently pronounced in both L- and P-band tomograms at

heights between $-15 \, \mathrm{m}$ to $-50 \, \mathrm{m}$, particularly in HV polarization (Fig. 6). Comparisons with Sounder data confirm the presence and height of the subsurface scattering layer, particularly at the locations of the radar-bright features (Sect. 4.3.1). For





radar-dark features, the subsurface scattering layer only becomes detectable after techniques are applied to remove dominant surface scattering in the tomographic analysis (Sect. 4.3.1). Modeling confirms the presence of this layer, although it appears weaker and is located deeper than in the radar-bright features (Sect. 4.4). Despite these differences, it is reasonable to assume

that the main subsurface layer is present across the entire test site. Therefore, the consistent presence of this subsurface scattering layer needs an exploration of its origin. Englacial channels are ruled out due to their typically localized nature (Catania et al., 2008). Additionally, a hypothesis that this layer originated from previous glacier zones is dismissed, as such layers would likely show slope and height variations from glacier movement and melting, which are not observed (Florentine et al., 2018). Our current hypothesis is that above this subsurface scattering layer, seasonal warming allows for temperature fluctuations,

while below lies impermeable ice (Sect. 4.3.2). The related transition in dielectric properties may cause the scattering. Variations in height of this layer may result from factors such as changes in the onset of impermeable ice, or variations in latent heat transmission (Florentine et al., 2018). Overall, this explanation would align with observed data and provides a coherent understanding of the subsurface scattering layer's characteristics and distribution across the test site, even though the available temperature measurements do not reach the depths of the observed subsurface layer (Sect. 4.3.2).

## 6 Conclusion

This study investigated the surface and subsurface characteristics of radar-bright and radar-dark features in the ablation zone of southwest Greenland, focusing on the Russell glacier. Using multi-frequency SAR data, we demonstrated that both radar-dark and radar-bright features are detectable in L- and P-band SAR images and can be effectively characterized using different advanced SAR techniques such as polarimetry, interferometry and tomography.

Radar-dark features were identified as low-backscatter surface scattering elements, typically convex-shaped and several hundred meters in size, often forming interconnected patterns. These features showed no correlation with surface topography or roughness, providing no evidence for drained or refrozen supraglacial lakes, although their shape suggests a possible connection. Their location and size exclude the 'dark zone' or cryoconite deposits as determining factors. Temporal analysis using ALOS-2 data showed that radar-dark features move with the glacier flow while remaining remarkably stable in shape and size,

despite the large surface melt.

In contrast, radar-bright features were characterized by high backscatter and volume scattering, with the main subsurface scattering layer located at heights of $-15\,\mathrm{m}$ to $-50\,\mathrm{m}$ in both L- and P-band. This layer's consistency over the complete testsite rules out englacial channels and formations from prior glacier zones as its origin. Glaciologically, this subsurface scattering layer likely forms due to seasonal temperature changes occurring above this layer, with impermeable ice lying beneath.

A preliminary glaciological interpretation suggests that radar-dark features are associated with the presence of a weathering crust, a porous layer that allows meltwater infiltration and refreezing. Through this process, residual liquid water content remains just below the ice surface during frozen conditions, leading to surface scattering and SAR signal attenuation. Therefore, the weathering crust best explains the characteristics of the radar-dark features, while radar-bright features likely represent bare, clean ice without a weathering crust. While our SAR and Sounder data has provided significant insights, further validation



using ground measurements such as ice core drillings, is essential to fully understand the formation processes of these features. Eventually, long-wavelength SAR could become a monitoring tool for weathering crust formation and near-surface meltwater storage in the ablation zone, contributing to the understanding of mass balance and runoff processes.

## Appendix A: Scattering Modeling Formulas and Parameters

The coherence magnitude for radar-bright features is modeled using a two-component approach that accounts for both surface and subsurface scattering contributions. The coherence magnitude as a function of polarization $w$ is given by:

$$\tilde{\gamma}_{\text{Vol}}(\omega) = e^{i\kappa_z z_0} \frac{\tilde{\gamma}_{Subsurface} + m(\omega)\tilde{\gamma}_{Surface}}{1 + m(\omega)} \tag{A1}$$

Here, $e^{i\kappa_z z_0}$ represents the phase shift due to the reference surface height $z_0$, which typically corresponds to the ice surface as defined by the reference DEM height. The surface-to-volume ratio $m(\omega)$ weights the surface and subsurface contributions, with its value depending on polarization, which affects the balance between surface and subsurface scattering. The surface component $\tilde{\gamma}_{Surface}$ is modeled using a rectangular function to account for surface roughness at the representative location (Cloude and Papathanassiou, 2003):

$$\tilde{\gamma}_{Surface} = e^{i\left(z_0 + \frac{h_v \kappa_z}{2}\right)} \frac{\sin\left(\frac{h_v \kappa_z}{2}\right)}{\frac{h_v \kappa_z}{2}} \tag{A2}$$

In this equation, the sinc function represents the Fourier transform of a rectangular structure function, incorporating $z_0$ for the reference surface height, $h_v$ as the height of the surface scattering layer, and $\kappa_z$ as the vertical wavenumber.

The subsurface component $\tilde{\gamma}_{Subsurface}$ is modeled using a Uniform Volume (UV) model, which assumes a uniform distribution of scatterers and captures diffuse scattering in subsurface ice layers (Zebker and Hoen, 2000; Fischer et al., 2020). The subsurface model is expressed as:

$$\tilde{\gamma}_{Subsurface} = e^{i\kappa_{zVol} z_{ul}} \frac{1}{1 + \frac{i d_{Pen}(\omega)\kappa_{zVol}}{2}} \tag{A3}$$

In this formula, $z_{ul}$ represents the upper limit of the subsurface scattering layer. The parameter $d_{Pen}(\omega)$, which varies with polarization, represents the penetration depth of the radar signal into this volume component.

The optimization process adjusts the parameters $m(\omega)$ and $d_{Pen}(\omega)$ with flexibility across different polarizations and frequency bands (Table A1). This approach enables the model to accurately fit the observed coherence magnitudes by effectively capturing the subsurface scattering structures within the ice. The height of the rectangular function for the surface scattering layer is $h_v = 2$ in all cases. The upper limit $z_{ul}$ for the UV model is set as $-15\,\text{m}$ for both L- and P-band consistently across different polarizations.

For radar-dark features, the scattering model consists of three components: a surface component, a weak subsurface scattering layer (not detected in Fig. 6d-e), and a deeper subsurface scattering layer (detected in Fig. 6e). The surface component is modeled using a rectangular function identical to that of radar-bright features, demonstrating the independence of surface roughness between the two features (Sect. 4.2). The weak subsurface layer is represented by a Dirac delta function at $-5\,\text{m}$,





**Table A1.** Modeling parameters for the radar-bright features, specifying the values for L- and P-band in both HH and HV polarizations

| Polarization | Surface-to-volume ratio $m(\omega)$ | | UV penetration depth $d_{Pen}(\omega)$ | |
| --- | --- | --- | --- | --- |
| | L-band | P-band | L-band | P-band |
| HH | 1.10 | 0.33 | 18 m | 15 m |
| HV | 0.13 | 0.08 | 12 m | 10 m |

accounting for additional scattering observed during the modeling approach. The deeper, main subsurface scattering layer is modeled using a UV model, with the upper limit adjusted to capture the deeper subsurface characteristics specific to radar-dark features. In this case, the upper limit for the UV model is set at $-48\,\mathrm{m}$ in both frequency bands. The modeling parameters for radar-dark features, including the surface-to-volume ratio $m(\omega)$, Dirac-to-volume ratio $m(\omega)$, and UV penetration depth $d_{Pen}(\omega)$, are detailed in Table A2.

**Table A2.** Modeling parameters for the radar-dark features, specifying the values for L- and P-band in both HH and HV polarizations

| Polarization | Surface-to-volume ratio $m(\omega)$ | | Dirac-to-volume ratio $m(\omega)$ | | UV penetration depth $d_{Pen}(\omega)$ | |
| --- | --- | --- | --- | --- | --- | --- |
| | L-band | P-band | L-band | P-band | L-band | P-band |
| HH | 25.00 | 15.71 | 1.25 | 0.71 | 20 m | 20 m |
| HV | 0.49 | 0.32 | 0.05 | 0.05 | 15 m | 30 m |





*Author contributions.* SS conducted the data analysis and wrote the manuscript. GF supervised the investigation of the ARCTIC15 campaign data and provided expertise in general SAR methods, particularly in modeling. MP supported with his expertise in SAR tomography. IH provided overall supervision and guided the content of the paper.

*Competing interests.* The authors declare that they have no conflict of interest.

*Acknowledgements.* I am profoundly grateful to the distinguished glaciologists who generously shared their expertise and resources to support this research. My sincere thanks go to Martin Lüthi of the University of Zurich (UZH), whose invaluable support and insights enriched the analysis of southwest Greenland. I am also deeply appreciative of Alun Hubbard from the Arctic University of Norway (UiT) for his expertise on the Russell Glacier, and Christoph Mayer of the Bayerische Akademie der Wissenschaften (BAdW), whose extensive knowledge of general glaciology greatly benefited this work.

I extend my gratitude to the dedicated teams involved in the ARCTIC15 campaign. Special thanks go to the DLR and ETH Zurich for the provision of essential data and resources, and to the flight, ground, and processing teams for their commitment and assistance.

This research was financially supported by the International Doctorate Program "Measuring and Modelling Mountain Glaciers and Ice Caps in a Changing Climate" ($M^3$OCCA).

Data acknowledgments include contributions from ENVEO and the PROMICE dataset. Some data used in this study were acquired by NASA's Operation IceBridge. ALOS-2/PALSAR-2 data were provided courtesy of JAXA, with appreciation to PI Jun-Su Kim, DLR. This research also makes use of Copernicus Sentinel-2 data provided by the European Space Agency (ESA). I extend my thanks to the providers and curators of these datasets, which were instrumental to this research.



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
