# Peer review of "Characterization of Ice Features in the Southwest Greenland Ablation Zone Using Multi-Modal SAR Data"

_EGUsphere, 2024_

## Referee Comment (RC1)

Review: Characterization of Ice Features in the Southwest Greenland
Ablation Zone Using Multi-Modal SAR Data

Sara-Patricia Schlenk et.al.

This study investigates ice features in the ablation zone of SW Greenland by using airborne SAR data in different frequency bands. SAR polarimetry, interferometry, tomography, and modeling is employed to characterize the features. Temporal stable low backscattering areas named as radar dark features are found, especially in the low frequency (L,P-band) domain which are interpreted as a weathering crust containing residual liquid water.  Radar bright features are characterized by volume scattering and the presence of a subsurface scattering layer is found along the transect.

In general, the paper is well written, clearly structured and the presented data set and methods to characterized ice features in the ablation zone and its interpretation is worth to be published in TC. Equations used are correct and figures are of good quality while mostly supporting the analysis. As the focus of the paper is a detailed interpretation of the data as well as to find a possible explanation of the origin of the radar dark features, I miss some more qualitative analysis of the data itself as well as the use of available external data sets.

General points:
The paper is a continuation of results presented in Parella et.al. (2021) and of the EUSAR conference paper of Pardini et.al. (2016) where the radar dark features were already described and Tomograms presented. I miss the reference to this paper as well as any discussion of those findings.
As mentioned above, I recommend to make use of the extensive data set of Operation Icebridge. CRESIS accumulation radar, which was used e.g. by Jullien et.al. (2023) to identify ice slab thickening, are available for 2010 and 2011 in the Russell glacier area and should be used in combination with the P-Band sounder data as it seems that the vertical resolution of the sounder data is not as good as the CRESIS accumulation radar. This could help to identify layers or inclusions in the upper 10m where the origin of the source of the radar dark features as well as to better explain the origin and regional extent of the deeper layer found in 20 to 50m depth.

All figures, where image data is shown are too small. I would recommend to use the full width of a page and high resolution to be able to zoom in.

When looking at the Radarsat compilation (20m C-Band, Joughin et.al. (2016)) one can already see the dark features (see figure 1 below - The blue lines show as an example some of CRESIS flight lines acquired on 8th April 2011)). To me it seems that some of those dark features are correlated with across flow surface crevasses. The circular features, which move with the ice as the authors explain, are situated in the middle of Russell glacier, and are maybe connected to a river network up slope and therefore are maybe relicts of drained lakes.

[Figure]

*Figure 1 RadarSat compilation with two CRESIS_accum profiles on top*

Could you please show an extra figure with a summer SENTINEL2 or Landsat image without snow cover as well as backscatter images of all three frequency bands, to be able to see surface crevasses (the airborne radar data is of higher resolution and should resolve the crevasses better than RadarSat). The crevasses would allow water to penetrate deeper than just the shallow weathering crust. In my opinion if water is present in the weathering crust you should also see dark features in X-Band as the weathering crust is usually less then 0.5m thick and the origin of the strong attenuation of L and P band seems to be deeper. How thick is the winter snow layer on top of the previous summer layer? Is there any information of the typical winter snow accumulation from the AWS nearby?

Specific remarks:

L85: ACTIC15 → ARCTIC15
Why don't you include C-Band as Parrela et.al. 2021?

Table1: Please specify the bandwidth of each system and add the wavelength lambda. This would help to understand to which surface roughness the different bands are sensitive.

L115 Fully focused SAR processing of sounder data. How was this achieved and is there a reference to be cited?

L125: Additional data sets:
As mentioned above, please make use of CRESIS accumulation radar for the top 80m

L223: Analysis of data
As mentioned above, please add backscatter images of all Bands and a summer optical image for crevasse detection.

Figure 3: Can you please mask the optical image in the same way as the radar images. This would make it easier to visually compare the images.

In your analysis you mention H and alpha. Please also show an extra figure with images of H, alpha, Anisotropy and add P and phi as images as well as in your analysis similar to Parella (2021). In addition, please generate the DEM gradient from your F-SAR DEM and include this as well. It would be interesting to see where surface depressions are located. Maybe a hillshade of the high-resolution DEM overlaid by a transparent DEM is also worth to show.

Table 4 and Table 5: Please include mean and standard deviation. Just showing the range is not really enough.

Can you please add a figure, were you show a scatter plot of H versus HH or HV Power as well as alpha vs power for all frequency band, as this seems to be correlated.
Please add a similar scatter plot with the phase center heights to illustrate the drastic change of phase center heights for L and P band in the radar dark areas.

Figure 4.
Again, too small. Why don't you show the DEM transparent on top of a hillshade. ATM data is not really telling me something here. To my opinion you can skip it. Insteade, please add also the DEM slope image or the slope image of the smoothed FSAR DEM which is used to detrend. Maybe surface depression can be seen, which are removed by the de-trended DEM.

Section 4.3
Figure 6. Is the sounder data depth scale converted to ice velocity?

Can you please create a similar figure as fig 6 but zoom in into the upper 10m or 20m ? As in this depth you expect the occurrence of the weathering crust containing liquid water, correct? Please extract H and /or HH,HV power along the flight track and plot this as extra subfigure in this new Figure. This would allow to directly align the dark features with the Tomograms.

For the Tomogram please don't use a spectral color scale. The scale should follow guidelines for color blind people . Please use the same color scale for the Tomograms and the Sounder radargram. Can you please add a similar figure for X and L Band (maybe in the appendix).

Can you please explain in more detail why this matrix filter to suppress surface scattering contribution picks up a layer in 30 to 50m depth, where no energy content was seen before? I'm wondering why the power level in subfigure f) after applying the matrix-filter is similar to the power level in areas where the deeper layer was seen before. Can you explain this?

In addition, please include CRESIS accumulation radargrams to compare with the p-band sounder data (see figure 2 below). Do you see a strong deep reflector in the same depth, and can you resolve strong near surface reflections in areas of the radar dark features? Is such a strong near surface reflector (if present) missing in the radar bright areas?

[Figure]

*Figure 2 Radargramm of CRESIS_accum (20110408_01_158). Trace 0 to 550 strong layer in roughly 10m - related to side lobe as surface return is very strong. From 550 onward dark reflection band similar to P-band sounder visible. This seem to be corelated with rough crevassed surface (surface clutter????)*

L355
I personally, don't agree that a thermal boundary is the main driver for the subsurface scattering layer in depth between 15 to 50m. The temperature wave is strongly suppressed with depth and should reach a value which corresponds to the annual mean surface temperature. This, is indicated in your plot, where the temperatures reaching -5°C in 10m depth.  How should a temperature boundary layer develop further down and how should this be responsible for a spatial varying reflection in 20 to 50 m depth? This idea needs more explanation!
I personally think, that this lower reflector could be related to radar clutter from near surface crevasses across track or the dark radar features itself, which the radar picks up from the side. At least this could explain the high spatial variability. Can you please make use of the dense grid of the CRESIS accumulation radar data to track this deeper layer and try to relate this to surface features across track. The CRESIS data would also allow to support your theory of the presence of a layer across the entire test site as stated in L485 (an example of the CRESIS data is given in figure 2 ).

Fig 8. And Fig 9
Please enlarge the labelling of axis.

Fig. 10
Please include in e,f,g the extent of the area as extra axis label right and top of each figure. This would allow to visually estimate the distance between the moved features and compare it to the average ice flow velocity.

L440 I think the water body needs to have a certain thickness to fully absorb the signal. This is related to the frequency. Even in areas of firn aquifer sometimes the bedrock is visible (see Horlings 2022).

L 432 and discussion later on

No link to surface roughness or topography …. I don't fully agree. See above – I think there is a link to crevasses as seen in Radarsat or maybe also to surface depressions. Those depression can be situated up slope. The circular features seem to be relicts of old lakes which were drained. As impurities are collected in lakes, they can later act (after lake drainage) to intensify the formation of a weathering crust allowing water to infiltrate the subsurface or enhance melting between grain boundaries. This argument you discuss later in L 465 ff but you mentioned that no direct evidence was found. For what evidence you were looking for and how do you rule this argument out?  I think you need to look in a spatially larger area.

Maybe its worth to have a look on an optical image during the melt season in the drainage catchment of Russell glacier. Where are lakes formed up slope and where do you see a denser river network? I could imagine that more water is present in the radar dark areas than in the bright areas during the melt season and maybe this kind of network is connected to lakes upslope. The band of radar dark features is also situated in the main trunk of Russell glacier where ice velocities are slightly larger. The glacier north of Russell (see figure above) shows even less backscatter in Radarsat more concentrated in the fast-flowing main trunk. Similar processes should be expected there to allow liquid water to penetrate and be present in the near surface layers. I fully agree that the dark features are related to liquid water content but I would not completely rule out a topographic driver.

Additional, as rocks are nearby more impurities can be expected in the ice matrix or between grains which were transported by water. Impurities enhance sub grain melting as well as the attenuation and might play a role here as well.

---

## Referee Comment (RC2)

Comments to the paper:
Characterization of Ice Features in the Southwest Greenland Ablation Zone Using Multi-Modal SAR Data

By S.P. Schlenk, G. Fischer, M. Pardini and I. Hajnsek

The paper presents an investigation based on airborne multi-frequency, polarimetric, interferometric and tomographic SAR data to characterize ice features detected in the ablation zone of the Greenland Ice Sheet. Additional data and results from other studies are used to support the analysis, including ALOS-2 data, P-band sounder measurements, Sentinel-2 optical data, lidar derived topographic data from NASA's IceBridge initiative and in-situ data of ice temperature.

The work focuses on radar-bright and radar-dark features which are more evident with increasing wavelength (i.e at L- and P-band), suggesting a link with subsurface features (weathering crust) and dynamics occurring at the study area.

General comments:

The paper has a clear structure and is easy to read. The analysis is presented in a clear way with an approach that tries to link the different aspects (polarimetry, interferometry, tomography) coherently.

There is potential to get additional insights by expanding the analysis to investigate for instance the dependency of PolSAR and InSAR signatures of the bright and dark features on the imaging geometry. In addition, the use of C-band data would also provide another piece of useful information.

The analysis is supported by a modelling effort which is able to explain the behavior of the PolInSAR coherence at L- and P-band, even though some assumptions, like the presence of specific layers not detectable in the sounder data and in the tomograms, are difficult to justify from a physical point of view.

Overall, the study presents new interesting elements for the interpretation of SAR data of land ice. The interpretation of the shallow subsurface scattering component with the presence of a weathering crust looks reasonable and it is supported by other studies. From a SAR modelling perspective, it suggests a possible source of scattering (to my knowledge) not yet considered in literature, which adds to the previously proposed sastrugi, oriented crevasse fields, volume scattering from air inclusions embedded into ice layers and icy inclusions embedded into firn.

Specific comments:

Eq. 7 :

Except the volumetric decorrelation, the other terms contributing to the InSAR coherence are not introduced/defined in the text.

Line 180 :

I suggest substituting the term 'subsurface ice elements' with 'subsurface scatterers' or 'subsurface scattering sources'.

Section 4.1

Here the authors start with the analysis of the PolSAR data which include X-, L- and P-band. As shown in Parrella et al. 2021 (JSTARS), this dataset includes also C-band acquisitions. It would be interesting to see also this data to have a more complete understanding of how the investigated radar-dark/bright features behave with frequency. InSAR and TomoSAR data should be available at C-band as well. Did the authors look at them?

Line 238:

I believe the authors here refer to Table 1 (instead of Table 2) when discussing about spatial resolution of P- and L-band data.

Table 4:

Here the authors report a summary of the value found with the polarimetric analysis at different frequencies.

My first point is that polarimetric signatures (including H and alpha angle) are typically influenced by the variation of incidence angle along the range direction. As both dark and bright features seem to be spread along the entire range direction in the SAR images, did the author assess whether the interval of observed H and alpha values is related to variation of incidence angle? Or are those values randomly occurring at different incidence angles? In general, it would be interesting to carry out an analysis of the polarimetric signatures with respect to the incidence angle for both bright and dark features. This might provide additional insights about the type of scattering mechanisms occurring in the two feature types, and support further the results of the InSAR and TomoSAR analysis.

My second point concerns the values observed at X-band for both H and alpha. At shorter wavelengths, I would expect overall higher values w.r.t. L- and P-band. I am a bit surprised to see that for radar-bright features, the lowest value of H and alpha is lower at X-band (0.2 and 0 degrees) than L- (0.5 and 20 deg) and P-band (0.6 and 30 degrees). Interestingly, the figures obtained for the radar-dark features are much more in line with the expected behavior. Do the authors have an interpretation of this phenomenon?

Table 5:

Also here, it would be interesting to know for which incidence angle are the estimated phase center height range representative, and if the authors observed any trend with the incidence angle.

Line 258-261:

I think that the limited penetration over the radar-dark features could also be related to the lack of effective scatterers deeper into the ice and not necessarily to a real shallow penetration (related to absorption). This is maybe an option to consider here.

Section 4.2.2, Line 284

Please replace 'albedo' with 'snow albedo'

Section 4.3.1

It is clear that the authors focus on P-band for the tomographic analysis since this provides an 'enhanced' sensitivity to subsurface scattering and deeper penetration. Anyway, it would be interesting to show and discuss also L-band tomograms and consistent with Section 4.4, where the InSAR modelling addresses both P and L-band measurements.

Line 376-378

'The general drop of coherence with increasing.... is typical for two scattering components with a certain vertical distance.' Please add a reference.

Figure 8 and 9

I miss here a brief discussion about the values of surface-to-volume ratio obtained to fit the data. Are they reasonable/explainable across polarizations (HH vs HV) and frequencies (L vs P)? The information reported in the appendix is explaining in more details the modelling approach and the obtained results, but it is not discussing them.

Please, also provide bigger images.

Section 5.1 and 5.2

In my opinion, these 2 paragraphs could be removed since they mostly summarize the findings of the analysis carried out in the previous sections.

Figures

Please expand Figures 1, 3, 5 to full page width and with better resolution. In some cases, it is difficult to observe the features and patterns discussed in the text (e.g. in Fig. 5).

---

## Author Comment (AC1)

**Review: Veit Helm, AWI**

Review: Characterization of Ice Features in the Southwest Greenland Ablation Zone Using Multi-Modal SAR Data

Sara-Patricia Schlenk et.al.

This study investigates ice features in the ablation zone of SW Greenland by using airborne SAR data in different frequency bands. SAR polarimetry, interferometry, tomography, and modeling is employed to characterize the features. Temporal stable low backscattering areas named as radar dark features are found, especially in the low frequency (L, P-band) domain which are interpreted as a weathering crust containing residual liquid water. Radar bright features are characterized by volume scattering and the presence of a subsurface scattering layer is found along the transect.

In general, the paper is well written, clearly structured and the presented data set and methods to characterized ice features in the ablation zone and its interpretation is worth to be published in TC. Equations used are correct and figures are of good quality while mostly supporting the analysis. As the focus of the paper is a detailed interpretation of the data as well as to find a possible explanation of the origin of the radar dark features, I miss some more qualitative analysis of the data itself as well as the use of available external data sets.

Dear Dr. Helm,
thank you very much for your thorough assessment of our manuscript and for your insightful comments. We truly appreciate you highlighting both the strengths and areas for improvement, which has been invaluable in helping us to enhance the paper. We have addressed your suggestions and have detailed the corresponding changes in response to each specific comment below.

**General points:**

The paper is a continuation of results presented in Parella et.al. (2021) and of the EUSAR conference paper of Pardini et.al. (2016) where the radar dark features were already described and Tomograms presented. I miss the reference to this paper as well as any discussion of those findings.

Thank you for this valuable comment. We agree that the connection to the previous work by Parella et al. (2021) and Pardini et al. (2016) could be articulated more clearly. We will revise the manuscript to explicitly reference these papers and discuss their findings in relation to the novel contributions presented in our current work.

As mentioned above, I recommend to make use of the extensive data set of Operation Icebridge. CRESIS accumulation radar, which was used e.g. by Jullien et.al. (2023) to identify ice slab thickening, are available for 2010 and 2011 in the Russell glacier area and should be used in combination with the P-Band sounder data as it seems that the vertical resolution of the sounder data is not as good as the CRESIS accumulation radar. This could help to identify layers or inclusions in the upper 10m where the origin of the source of the radar dark features as well as to better explain the origin and regional extent of the deeper layer found in 20 to 50m depth.

We appreciate you making us revisit the CRESIS accumulation radar data from Operation Icebridge, as we looked into it in the past but not anymore with our current understanding.
We did investigate the CRESIS dataset and observed similar patterns to those found in our F-SAR Sounder data, see the comparison of the two sounders at the related specific remark. Due to the very similar information content, we've decided to proceed with our own sounder data for this particular study, because of the closer temporal correlation of the Sounder and SAR data.

However, we certainly recognize the value of the CRESIS data. We plan to keep it in mind for future research, particularly for comparative studies that could benefit from a broader temporal and spatial coverage.

All figures, where image data is shown are too small. I would recommend to use the full width of a page and high resolution to be able to zoom in.

Thank you for pointing this out. We will ensure that all figures displaying image data are appropriately sized and are presented in high resolution to allow for detailed examination and zooming.

When looking at the Radarsat compilation (20m C-Band, Joughin et.al. (2016)) one can already see the dark features (see figure 1 below - The blue lines show as an example some of CRESIS flight lines acquired on 8th April 2011)). To me it seems that some of those dark features are correlated with across flow surface crevasses. The circular features, which move with the ice as the authors explain, are situated in the middle of Russell glacier, and are maybe connected to a river network up slope and therefore are maybe relicts of drained lakes.

 We agree that these features are indeed visible in C-band, which aligns well with the behavior observed in our own data.
Regarding a clear correlation with across-flow surface crevasses, we did not find a consistent pattern. While some radar-dark features appear in highly crevassed areas, others are present in relatively smooth regions, suggesting that crevassing alone is not the sole or consistent driver for their formation across our study site.
However, your hypothesis about a connection to relicts of old, drained supraglacial lakes is indeed plausible and supported by our findings. The distinct circular shapes of some of these radar-dark features support this idea. A former lake bed can accumulate sediments or other impurities over time, which can enhance the absorption of solar radiation, leading to increased local melting. This process can potentially maintain a higher liquid water content within the ice below the former lake, even after drainage. Such areas, with increased liquid water content, would appear radar-dark due to strong C-band attenuation.
We also wish to clarify that we are referring to old relicts of supraglacial lakes, not drained lakes from the most recent summer season. Our previous analysis, based on current topography and surface roughness, allowed us to rule out recent lake drainage as the primary explanation for all observed radar-dark features (see Sect. 4.2). While the direct connection to an upslope river network was not extensively researched in the scope of this paper, we agree that this is a compelling aspect for further investigation.

[Figure]

*Figure 1 RadarSat compilation with two CRESIS_accum profiles on top*

Could you please show an extra figure with a summer SENTINEL2 or Landsat image without snow cover as well as backscatter images of all three frequency bands, to be able to see surface crevasses (the airborne radar data is of higher resolution and should resolve the crevasses better than RadarSat). The crevasses would allow water to penetrate deeper than just the shallow weathering crust. In my opinion if water is present in the weathering crust you should also see dark features in X-Band as the weathering crust is usually less than 0.5m thick and the origin of the strong attenuation of L and P band seems to be deeper.

Thank you for your question! We plotted the backscatter images in all three frequency bands together with a Sentinel-2 image (see "Specific remarks"). Based on our analysis:

- Connection to surface crevasses: Despite the high resolution of our data, we couldn't detect a direct or consistent connection between the radar-dark features and surface crevasses across the entire test site. While crevasses are present, the radar-dark features appear in areas with both high crevasse density and relatively smooth ice, indicating that crevassing is not the sole or consistent cause of these features (see Section 4.2). Furthermore, given the winter acquisition, we don't believe there is free water present either deeper in the crevasses or directly below the surface; rather, our interpretation suggests a slightly higher liquid water content distributed within the ice matrix in the radar-dark features compared to the surrounding radar-bright features.

- Distinctness in X-band vs. L- and P-band: You are correct to inquire why the radar-dark features are less clearly visible in X-band compared to L- and P-band. The primary reason is that our test site has quite a high surface roughness. With its small wavelength, the X-band backscatter is largely dominated by this surface roughness and is less influenced by the variations in dielectric properties within the ice. You can see this quite clearly by comparing the Sentinel-2 image with the X-band RGB Pauli representation in Figure 3 of our paper, both showing the same features. Unlike X-band, L- and P-band, with their longer wavelengths, can penetrate deeper and are more sensitive to the dielectric properties of the glacier.

- Attenuation origin: Our hypothesis is that the origin of the strong attenuation for L- and P-band at the radar-dark features is indeed the shallow weathering crust, which we interpret as containing a small amount of liquid water content distributed within the ice matrix. This attenuation, occurring within this shallow layer, is significant enough to effectively mask

deeper signals, making it difficult to resolve additional subsurface scatterers without specialized processing such as our matrix filter (see Section 4.3.1). Thus, the strong attenuation at L- and P-band is attributed to this shallow layer, not necessarily from a deeper source.

How thick is the winter snow layer on top of the previous summer layer? Is there any information of the typical winter snow accumulation from the AWS nearby?

The typical snow height around the testsite is 0.5 to 1.5m (from PROMICE Data for KAN L station). We will add this information in Section 4.1, where the optical Sentinel-2 image is shown.

**Specific remarks:**

L85: ACTIC15 → ARCTIC15

Thank you for catching that typo! Corrected.

Why don't you include C-Band as Parrela et al. 2021?

Thank you for raising the point about including C-band data, especially given its relevance in Parrella et al. (2021) and for satellite missions like Sentinel-1 and RadarSAT.
We did indeed analyze C-band data during our study. However, given the already considerable length of the paper and our desire to maintain focus, we made the decision to concentrate on X-band for surface analysis and L-band and P-band for subsurface analysis. We felt these chosen bands most effectively demonstrate the range of penetration and sensitivity to their respective scattering components (surface and subsurface features) that are central to our paper's narrative. While C-band can offer valuable information, it appeared as "intermediate" case between X- and L-band in all our analyses. Thus, we believe the current selection provides a comprehensive understanding within the scope of this manuscript.

Table1: Please specify the bandwidth of each system and add the wavelength lambda. This would help to understand to which surface roughness the different bands are sensitive.

Thanks for the suggestion! We'll include the bandwidth and wavelength for each system in Table 1. This will indeed help clarify the sensitivity of different bands to surface roughness.

L115 Fully focused SAR processing of sounder data. How was this achieved and is there a reference to be cited?

Thank you for raising this important point regarding the SAR processing of the sounder data.
We have already cited the only relevant conference paper that describes aspects of this specific processing chain by the DLR: Scheiber et al., 2008. We will merge the two sentences to make it clearer that the reference is also for the SAR processing. However, only some aspects of the along-track SAR focusing are explicitly described in this reference, with further reference to existing SAR algorithms. Currently, there are no other dedicated publications detailing the specific methodology for F-SAR sounder data processing beyond what is referenced.

L125: Additional data sets:
As mentioned above, please make use of CRESIS accumulation radar for the top 80m

We deeply appreciate your continued recommendation to consider the CRESIS accumulation radar data. This is a valuable suggestion, and we did thoroughly investigate this dataset, finding patterns quite similar to those in our F-SAR Sounder data. For the scope of this paper, we've decided to prioritize our F-SAR data due to its temporal proximity.

L223: Analysis of data
As mentioned above, please add backscatter images of all Bands and a summer optical image for crevasse detection .

Thank you for your suggestion to add a figure with a summer optical image and backscatter images of all three frequency bands for crevasse detection. For your reference, we have plotted a Sentinel-2 summer scene along with X-, L-, and P-band HV backscatter images below in this response document. However, to maintain the focus and conciseness of the main manuscript, and considering that the P-band backscatter is already shown in Figure 1 and the RGB representation of all frequency bands in Figure 3, we have opted not to include this specific combination as a new figure in the paper itself, as it does not introduce substantially new information beyond what is already presented or discussed.

[Figure]

Figure 3: Can you please mask the optical image in the same way as the radar images. This would make it easier to visually compare the images.

Thank you for your valuable suggestion regarding masking the optical image in Figure 3 to match the radar images. We appreciate the intention to facilitate visual comparison.
However, due to inherent differences in the acquisition and processing steps for each frequency band, the masks for X-, L-, and P-band are distinct. These unique masks are important as they allow us to present the maximum possible data coverage for each frequency band across the testsite. Therefore, we propose to geocode the Pauli RGB images, making a precise visual comparison more straightforward despite the differing masks.

In your analysis you mention H and alpha. Please also show an extra figure with images of H, alpha, Anisotropy and add P and phi as images as well as in your analysis similar to Parella (2021).

Thank you for the detailed feedback regarding the polarimetric parameters and the suggestion to include figures for Anisotropy, P, and phi, similar to Parrella (2021).

In our paper, the primary focus for using Entropy (H) and Mean Alpha Angle (α) in our polarimetric analysis is to effectively distinguish and characterize the dominant scattering mechanisms present in the ice features. These two parameters are fundamental for identifying surface, volume, and dihedral scattering, which is crucial for understanding the properties of the radar-dark and radar-bright features, providing a clear and interpretable framework for categorizing the observed radar signatures (surface scattering for radar-dark features and volume scattering for radar-bright features).

While other polarimetric parameters like Anisotropy, P (co-polarization power ratio), and phi (co-pol phase difference) can offer more detailed insights into scattering processes, we believe that H and α are sufficient for the specific characterization goals and the overall scope of this study. To maintain clarity and focus within the current manuscript, we will therefore not include additional figures for these parameters.

The Entropy and Alpha Maps for X-, L-, and P-band of the testsite, shown below, provide valuable visual support for our polarimetric analysis and will be included in the appendix of the paper.

[Figure]

[Figure]

In addition, please generate the DEM gradient from your F-SAR DEM and include this as well. It would be interesting to see where surface depressions are located. Maybe a hillshade of the high-resolution DEM overlaid by a transparent DEM is also worth to show.

We have indeed already generated the DEM gradient for our F-SAR data, as well as for the ArcticDEM. We believe this data will be very helpful in identifying surface depressions and providing additional context. In the Preprint Answer Document, we suggested already to include our DEM in the appendix. We will add a corresponding reference to the full F-SAR DEM figure at the end of section 4.2.1.

[Figure]

Table 4 and Table 5: Please include mean and standard deviation. Just showing the range is not really enough.

Thank you for your valuable feedback on Tables 4 and 5, and for suggesting the inclusion of mean and standard deviation. We agree that providing only the range does not offer a comprehensive understanding of the entire test site's characteristics. Therefore, we will include the mean and standard deviation for both tables:

- For Table 4 (polarimetric analysis of Entropy and Alpha): These statistics will be added to the table. Additionally, their corresponding maps for the three frequency bands which will be placed in the Appendix will provide a good impression of the scattering mechanisms present.
- For Table 5 (Phase Center Depths): We will also include the mean and standard deviation for the phase center depth measurements.

Figure 4.
Again, too small. Why don't you show the DEM transparent on top of a hillshade. ATM data is not really telling me something here. To my opinion you can skip it. Instead, please add also the DEM slope image or the slope image of the smoothed FSAR DEM which is used to detrend. Maybe surface depression can be seen, which are removed by the de-trended DEM.

Thank you for your valuable feedback on Figure 4. We will ensure Figure 4 is presented in a larger format for improved readability. We added a standard DEM plot as Figure 4c to further illustrate the local topography. While we understand your idea of a connection to surface features, our thorough checks on the DEM data have consistently shown no direct connection between the observed features and local topography or surface roughness. Therefore, the here added DEM of the sample area is unlikely to reveal further relevant insights into the radar-dark and radar-bright features. For a comprehensive overview of the test site's overall topography, we will include a DEM and its de-trended version in the Appendix.

[Figure]

Section 4.3
Figure 6. Is the sounder data depth scale converted to ice velocity?

Yes, the sounder data depth scale in Figure 6 is converted, assuming an ice dielectric content of 3.15. We will add "All heights are scaled assuming a permittivity of 3.15" to the figure caption.

Can you please create a similar figure as fig 6 but zoom in into the upper 10m or 20m? As in this depth you expect the occurrence of the weathering crust containing liquid water, correct?

Thank you for your suggestion to create a zoomed-in figure of the upper 10-20m of the Sounder data. We appreciate your interest in focusing on the shallow depths where the weathering crust containing liquid water is expected. However, the primary purpose of Figure 6 is to illustrate the full depth and location of all significant scatterers present in the ice, which are also found below the 10-20m range. Instead, to provide a more targeted understanding of the subsurface scattering layer, we present another figure presenting a comparison of L- and P-band tomograms in both HH and HV polarization over a smaller, representative sample area of 1km in azimuth. This figure includes Pauli RGB representations in L-band (left) and P-band (right) with a white transect line indicating the tomogram locations on top. This figure below demonstrates:

- In general, HH polarization exhibits more pronounced scattering at the surface, while HV polarization reveals more prominent subsurface scattering, a trend observed for both frequencies.
- Critically, in L-band, the subsurface scattering layer appears as a distinct, clear line, suggesting limited penetration below this layer. In contrast, P-band clearly penetrates into the medium below and therefore the layer appears wider and has more variation along the depth axis.

[Figure]

While we appreciate your suggestion, its description and interpretation are very similar to what is already presented in Section 4.3.1. Therefore, to avoid redundancy and maintain the focus and conciseness of the main manuscript, we propose not to include this plot in the paper itself.

Please extract H and /or HH, HV power along the flight track and plot this as extra subfigure in this new Figure. This would allow to directly align the dark features with the Tomograms.

Thank you for your request to extract and plot H along the flight track as an extra subfigure. For your reference, we have plotted this Entropy Line Plot for the Sounder track below. While we appreciate your suggestion for this visualization, to maintain the focus and conciseness of the main manuscript, we have chosen not to include this plot in the paper itself.

[Figure]

For the Tomogram please don't use a spectral color scale. The scale should follow guidelines for color blind people. Please use the same color scale for the Tomograms and the Sounder radargram.

Thank you for your valuable suggestion concerning the color scale for the tomograms. We recognize the importance of employing colorblind-friendly palettes and appreciate your thoughtful consideration. While we acknowledge the benefits of such scales, the spectral color scale currently used for tomograms represents a standard presentation within the SAR community. To maintain continuity with established practices and facilitate comparison with prior research, we would prefer to retain this convention.

Can you please add a similar figure for X and L Band (maybe in the appendix)?

Thank you for your suggestion to include tomograms for X- and L-band.
Regarding an X-band tomogram: We believe that an X-band tomogram would not provide significant additional insight for the main paper. Due to the high sensitivity of X-band to surface scattering and the penetration depth limitation, the resulting tomogram would primarily depict the surface, and the accuracy of subsurface features at this penetration scale is questionable depending on the precise adjustment of the X-band DEM. However, if you are interested in a visual example of an X-band tomogram, showing basically only the glacier surface, we refer you to the conference paper by G. Parella (2019): "Interpretation of polarimetric and tomographic signatures from glacier subsurface: The K-Transect case study."
Regarding an L-band tomogram: For your reference, we have included a comparison of L- and P-band tomograms of a smaller testsite above. These tomograms show that L-band returns very similar scatterers to P-band, though to a shallower depth, as the signal experiences quicker attenuation. This results in a more concise scattering layer. While these plots illustrate valuable insights into L-band behavior, its interpretation closely is more related to frequency-dependent penetration, then new findings. Therefore, to maintain the focus and conciseness of the main paper, we propose not to include this plot in the paper itself.

Can you please explain in more detail why this matrix filter to suppress surface scattering contribution picks up a layer in 30 to 50m depth, where no energy content was seen before? I'm wondering why the power level in subfigure f) after applying the matrix-filter is similar to the power level in areas where the deeper layer was seen before. Can you explain this?

This is an excellent point for clarification.
The invisibility of this deeper layer before filtering is primarily due to the presentation of the color scale. Initially, the stronger surface return dominates the display, completely masking the much weaker subsurface signals. Once the surface scattering component is suppressed by the filter, the remaining, albeit faint, subsurface signals can then be scaled to fill the available dynamic range of the color scale, making them visible.
This plot serves to qualitatively demonstrate that a signal can be detected below the surface—even in conditions where a stronger signal return (e.g., from a certain amount of water on the surface or nearby) would otherwise obscure it.

In addition, please include CRESIS accumulation radargrams to compare with the p-band sounder data (see figure 2 below). Do you see a strong deep reflector in the same depth, and can you resolve strong near surface reflections in areas of the radar dark features? Is such a strong near surface reflector (if present) missing in the radar bright areas?

Thank you for your suggestion to include CRESIS accumulation radargrams and for providing the specific example. While we couldn't locate the exact CRESIS track from 08.04.2011 for direct overlay

with our test site, we did compare our Sounder data with other available CRESIS tracks. Here's what we found regarding the reflectors:

- Deep Reflector:
  - CRESIS data shows similar patterns for a deep layer as observed in our F-SAR Sounder data, suggesting a consistent subsurface feature across different radar systems. Therefore, we have decided to stay with only F-SAR Sounder data in the paper due to its better temporal fit with our other SAR measurements.
  - The "dark reflection band" you describe aligns with what we interpret as our subsurface scattering layer, representing the transition between the low-backscatter layer above and the higher-backscatter area below. This "band" is generally located deeper for the radar-dark features and higher for the radar-dark features (see Tomographic analysis and Modelling approach).

- Near-Surface Reflections:
  - In the CRESIS data we examined, no clear or consistent difference in backscatter for near-surface reflections was evident between areas corresponding to radar-dark and radar-bright features.
  - While our F-SAR Sounder data showed a potential for subtle surface reflection differences (slightly lower for radar-bright, slightly higher for radar-dark), this was not consistently observed for every feature. Therefore, this specific surface signature was inconclusive as a general differentiator.
  - We also could not detect a direct connection between these features and rough, crevassed surfaces (see surface roughness and topography discussion).

[Figure]

[Figure]

*Figure 2 Radargramm of CRESIS_accum (20110408_01_158). Trace 0 to 550 strong layer in roughly 10m - related to side lobe as surface return is very strong. From 550 onward dark reflection band similar to P-band sounder visible. This seem to be corelated with rough crevassed surface (surface clutter????)*

L355

I personally, don't agree that a thermal boundary is the main driver for the subsurface scattering layer in depth between 15 to 50m. The temperature wave is strongly suppressed with depth and should reach a value which corresponds to the annual mean surface temperature. This, is indicated in your plot, where the temperatures reaching -5°C in 10m depth. How should a temperature boundary layer develop further down and how should this be responsible for a spatial varying reflection in 20 to 50 m depth? This idea needs more explanation!

Thank you for raising this very important point and for challenging our interpretation of the deeper subsurface scattering layer. You're right that a temperature boundary layer is currently a theory without definitive proof. Our interpretation stems from the layer's consistent appearance across multiple F-SAR Sounder tracks and its corroboration by our extensive tomographic analysis (also by looking at many tomograms at different locations in the testsite). We're confident it's not surface clutter or an artifact, as it's seen consistently with both side-looking SAR from different headings and nadir-looking Sounder geometries. This broad spatial distribution led us to consider a widespread phenomenon like a temperature boundary. We agree that the exact details of this theory require further dedicated investigation with additional ground measurements in future research, and we will clarify its theoretical nature in the manuscript in Section 5.3.2.

I personally think, that this lower reflector could be related to radar clutter from near surface crevasses across track or the dark radar features itself, which the radar picks up from the side. At least this could explain the high spatial variability. Can you please make use of the dense grid of the CRESIS accumulation radar data to track this deeper layer and try to relate this to surface features across track? The CRESIS data would also allow to support your theory of the presence of a layer across the entire test site as stated in L485 (an example of the CRESIS data is given in figure 2).

We appreciate you considering the CRESIS accumulation radar data to track this deeper layer and relate it to surface features.
We did consider using the CRESIS data for this purpose. However, we have multiple F-SAR Sounder tracks across our test site, all of which show consistent results for this deeper layer. Further, we computed also several additional tomograms (not shown in the paper) across the test site to confirm the consistent presence of this deeper layer. Because our own dataset provides sufficient coverage and consistency for our current analysis, we've decided to focus on our F-SAR Sounder and TomoSAR data.

Fig 8. And Fig 9
Please enlarge the labelling of axis.

No problem at all! We've enlarged the labeling of the axes in both Figure 8 and Figure 9 to improve readability.

Fig. 10
Please include in e,f,g the extent of the area as extra axis label right and top of each figure. This would allow to visually estimate the distance between the moved features and compare it to the average ice flow velocity.

Good suggestion! We'll add the extent of the area as extra axis labels on the right and top of figures e, f, and g in Figure 10.

L440 I think the water body needs to have a certain thickness to fully absorb the signal. This is related to the frequency. Even in areas of firn aquifer sometimes the bedrock is visible (see Horlings 2022).

You are absolutely correct that the degree of signal absorption is indeed frequency-dependent. Regarding the reference to Horlings (2022) and bedrock visibility, it is important to note a key difference in methodology. Horlings (2022) uses a very low-frequency Multichannel Coherent Radar Depth Sounder (RDS) system, which operates at a significantly lower frequency than our P-band Sounder mode of F-SAR, to detect the bedrock.

Crucially, we still refer to signal attenuation, not complete absorption, as we detect some signal below the radar-dark features (see Section 4.3.1). Therefore, our interpretation suggests that significant attenuation does not necessarily require a substantial, contiguous "water body." Instead, even small amounts of distributed liquid water content, such as residual moisture within the ice structure, can be sufficient to cause significant attenuation of the radar signal at these frequencies.

L 432 and discussion later on

No link to surface roughness or topography …. I don't fully agree. See above – I think there is a link to crevasses as seen in Radarsat or maybe also to surface depressions. Those depression can be situated up slope. The circular features seem to be relicts of old lakes which were drained. As impurities are collected in lakes, they can later act (after lake drainage) to intensify the formation of a weathering crust allowing water to infiltrate the subsurface or enhance melting between grain boundaries. This argument you discuss later in L 465 ff but you mentioned that no direct evidence was found. For what evidence you were looking for and how do you rule this argument out? I think you need to look in a spatially larger area.

Maybe it's worth to have a look on an optical image during the melt season in the drainage catchment of Russell glacier. Where are lakes formed up slope and where do you see a denser river network? I could imagine that more water is present in the radar dark areas than in the bright areas during the melt season and maybe this kind of network is connected to lakes upslope. The band of radar dark features is also situated in the main trunk of Russell glacier where ice velocities are slightly larger. The glacier north of Russell (see figure above) shows even less backscatter in Radarsat more concentrated in the fast-flowing main trunk.

Similar processes should be expected there to allow liquid water to penetrate and be present in the near surface layers. I fully agree that the dark features are related to liquid water content but I would not completely rule out a topographic driver.

Additional, as rocks are nearby more impurities can be expected in the ice matrix or between grains which were transported by water. Impurities enhance sub grain melting as well as the attenuation and might play a role here as well.

Thank you for your extensive and insightful feedback on the potential links between radar-dark features, surface roughness, topography, and impurities, especially your hypothesis about relicts of old lakes. We truly appreciate this detailed analysis.

Maybe our statement about "no link to surface roughness or topography" needs clarification. We meant that within our test site, we found no immediate, consistent, and directly observable correlation between current surface roughness/topography and the radar-dark features across their full extent. We observed radar-dark features in both depressions and elevated areas, as well as in areas with varying surface roughness, which led us to conclude there wasn't a simple, direct topographic or roughness-based explanation for all observed features. This was based on detailed analysis of our acquired data.

However, we entirely agree with your argument that the circular features strongly suggest a connection to relics of old, drained supraglacial lakes. This is a highly plausible origin. When we previously mentioned "no direct evidence was found," we were specifically looking for immediate indicators like smoother surfaces (for refrozen lakes) or clear depressions (for recently drained lakes) that directly matched the radar-dark features, but as you rightly pointed out, old relics might not show such obvious signs.

Your suggestion that impurities collected in these past lakes could intensify weathering and liquid water infiltration is a very strong argument, and we find it aligns well with our weathering crust hypothesis. We acknowledge that looking at a spatially larger area and during melt season might reveal more connections to upslope lake formation and river networks, and this is indeed a valuable direction for future studies. However, we don't believe that they could solely explain the extreme differences in the analysis.

In summary, we concur that the relict lake idea is a very strong candidate for the origin of some radar-dark features. We will refine our discussion in L432ff to reflect that while a direct, recent topographical driver might not be immediately apparent, the historical topographic context (like old lakes) is a highly probable contributing factor.

---

## Author Comment (AC2)

**Review: Giuseppe Parrella**

Comments to the paper:

Characterization of Ice Features in the Southwest Greenland Ablation Zone Using Multi-Modal SAR Data

By S.P. Schlenk, G. Fischer, M. Pardini and I. Hajnsek

The paper presents an investigation based on airborne multi-frequency, polarimetric, interferometric and tomographic SAR data to characterize ice features detected in the ablation zone of the Greenland Ice Sheet. Additional data and results from other studies are used to support the analysis, including ALOS-2 data, P-band sounder measurements, Sentinel-2 optical data, lidar derived topographic data from NASA's IceBridge initiative and in-situ data of ice temperature.

The work focuses on radar-bright and radar-dark features which are more evident with increasing wavelength (i.e at L- and P-band), suggesting a link with subsurface features (weathering crust) and dynamics occurring at the study area.

Dear Dr. Parella,

Thank you for your valuable feedback! Since you have great insight into the dataset, test site, and methods, your input is very much appreciated by us. We truly value your thorough assessment of our manuscript and your insightful comments, which have been helping us to enhance the paper. We have addressed your suggestions and have detailed the corresponding changes in response to each specific comment below.

**General comments:**

The paper has a clear structure and is easy to read. The analysis is presented in a clear way with an approach that tries to link the different aspects (polarimetry, interferometry, tomography) coherently.

There is potential to get additional insights by expanding the analysis to investigate for instance the dependency of PolSAR and InSAR signatures of the bright and dark features on the imaging geometry. In addition, the use of C-band data would also provide another piece of useful information.

The analysis is supported by a modelling effort which is able to explain the behavior of the PolInSAR coherence at L- and P-band, even though some assumptions, like the presence of specific layers not detectable in the sounder data and in the tomograms, are difficult to justify from a physical point of view.

Overall, the study presents new interesting elements for the interpretation of SAR data of land ice. The interpretation of the shallow subsurface scattering component with the presence of a weathering crust looks reasonable and it is supported by other studies. From a SAR modelling perspective, it suggests a possible source of scattering (to my knowledge) not yet considered in literature, which adds to the previously proposed sastrugi, oriented crevasse fields, volume scattering from air inclusions embedded into ice layers and icy inclusions embedded into firn.

**Specific comments:**

Eq. 7:
Except the volumetric decorrelation, the other terms contributing to the InSAR coherence are not introduced/defined in the text.

Thank you for pointing out that the other terms contributing to the InSAR coherence in Eq. 7 are not explicitly introduced or defined.
The volume decorrelation is the primary focus and the most significant decorrelation source for our analysis in this paper. However, we understand that readers might benefit from additional context on the other terms. Therefore, we will add an additional citation directly after introducing these terms, directing interested readers to a relevant source for their definitions and further explanation.

Line 180:
I suggest substituting the term 'subsurface ice elements' with 'subsurface scatterers' or 'subsurface scattering sources'.

Good idea! We'll replace the term.

Section 4.1
Here the authors start with the analysis of the PolSAR data which include X-, L- and P-band. As shown in Parrella et al. 2021 (JSTARS), this dataset includes also C-band acquisitions. It would be interesting to see also this data to have a more complete understanding of how the investigated radar-dark/bright features behave with frequency. InSAR and TomoSAR data should be available at C-band as well. Did the authors look at them?

Thank you for raising the point about including C-band data, especially given its relevance in Parrella et al. (2021) and for satellite missions like Sentinel-1 and RadarSAT.
We did indeed analyze C-band data during our study. However, given the already considerable length of the paper and our desire to maintain focus, we made the decision to concentrate on X-band for surface analysis and L-band and P-band for subsurface analysis. We felt these chosen bands most effectively demonstrate the range of penetration and sensitivity to their respective scattering components (surface and subsurface features) that are central to our paper's narrative. While C-band is generally also interesting, it represents an intermediate case between X- and L-band in our analyses. Thus, we believe the current selection provides a comprehensive understanding within the scope of this manuscript.

Line 238:
I believe the authors here refer to Table 1 (instead of Table 2) when discussing about spatial resolution of P- and L-band data.

Thank you for pointing that out!

Table 4:
Here the authors report a summary of the value found with the polarimetric analysis at different frequencies.

My first point is that polarimetric signatures (including H and alpha angle) are typically influenced by the variation of incidence angle along the range direction. As both dark and bright features seem to be spread along the entire range direction in the SAR images, did the author assess whether the interval of observed H and alpha values is related to variation of incidence angle? Or are those values randomly occurring at different incidence angles? In general, it would be interesting to carry out an analysis of the polarimetric signatures with respect to the incidence angle for both bright and dark features. This might provide additional insights about the type of scattering mechanisms occurring in the two feature types, and support further the results of the InSAR and TomoSAR analysis.

Thank you for this insightful point regarding the influence of incidence angle on polarimetric signatures, specifically Entropy (H) and Mean Alpha Angle (α). You are absolutely correct that these parameters are typically affected by incidence angle variations.

Our findings show a small systematic variation of both H and α with the incidence angle across the range. Here, we observe lower H and α at steeper incidence angles (near range), transitioning to slightly higher values as the incidence angle increases (towards far range). However, as can be seen in the provided plot, visually, Entropy and Alpha Angle stay relatively stable across the variation in incidence angles.

This indicates that while a systematic trend exists, the primary polarimetric characteristics used to distinguish the radar-dark and radar-bright features show smaller incidence angle variations than the general spatial variations (e.g. comparing different dark features). Therefore, we prefer to not include a dedicated analysis over incidence angle. This stability further strengthens the robustness of our interpretations regarding the dominant scattering mechanisms (see P band plots below).

[Figure]

[Figure]

My second point concerns the values observed at X-band for both H and alpha. At shorter wavelengths, I would expect overall higher values w.r.t. L- and P-band. I am a bit surprised to see that for radar-bright features, the lowest value of H and alpha is lower at X-band (0.2 and 0 degrees) than L- (0.5 and 20 deg) and P-band (0.6 and 30 degrees). Interestingly, the figures obtained for the radar-dark features are much more in line with the expected behavior. Do the authors have an interpretation of this phenomenon?

Thank you for this observation regarding the unexpected X-band H and Alpha values for radar-bright features. We agree this needs careful consideration.
We acknowledge that the initial parameters were derived from a sample area, which may not have been fully representative. To address this, we are re-evaluating these parameters using the entire test site, and will include the mean and standard deviation for H and Alpha in the revised manuscript.

Table 5:
Also, here, it would be interesting to know for which incidence angle are the estimated phase center height range representative, and if the authors observed any trend with the incidence angle.

Thank you for this question regarding the incidence angle for the estimated phase center height range in Table 5, and for asking about any observed trends. The range was indeed chosen for a sample area with an incidence angle between 30-40°.
Regarding the trend, our observations align with the general principle that there is less effective penetration with a higher incidence angle. This is because at higher incidence angles, the radar signal travels a longer path through the ice column to reach a given vertical depth, leading to increased attenuation due to both absorption and volume scattering. This effectively limits the vertical penetration capability of the radar. However, the relative difference in penetration between radar-bright and radar-dark stay the same across incidence angle, which is the important message from Table 5.

Line 258-261:
I think that the limited penetration over the radar-dark features could also be related to the lack of effective scatterers deeper into the ice and not necessarily to a real shallow penetration (related to absorption). This is maybe an option to consider here.

Thank you for raising the excellent point that limited penetration over radar-dark features could stem from a lack of effective scatterers deeper in the ice. This is a very valuable alternative to consider. However, our observations lead us to favor the attenuation hypothesis:

- Detection of Subsurface Scatterers Post-Filtering: After applying our matrix filter for surface cancellation, we are able to detect scatterers below these radar-dark features. This indicates that scatterers are indeed present; they were simply masked by the overwhelming signal attenuation closer to the surface.
- Consistency Across Multiple Geometries: It is highly unlikely that there would be a genuine absence of scatterers given our diverse acquisition geometries. We observe this shallow penetration consistently from side-looking SAR in two different headings and from nadir-looking Sounder data. The uniformity of this observation across multiple perspectives strongly suggests a widespread attenuating medium, rather than a spatially variable lack of scatterers.

Therefore, while we appreciate considering the alternative, our data robustly support the interpretation that attenuation due to liquid water in the weathering crust is the primary cause of the limited penetration.

Section 4.2.2, Line 284
Please replace 'albedo' with 'snow albedo'

Sure, we'll replace "albedo" with "snow albedo".

Section 4.3.1
It is clear that the authors focus on P-band for the tomographic analysis since this provides an 'enhanced' sensitivity to subsurface scattering and deeper penetration. Anyway, it would be interesting to show and discuss also L-band tomograms and consistent with Section 4.4, where the InSAR modelling addresses both P and L-band measurements.

Thank you for this insightful suggestion. We entirely agree that incorporating L-band tomograms would provide a more complete picture and align well with our multi-frequency InSAR modeling presented in Section 4.4. Therefore, we propose to include an L-band HV tomogram over the same transect as our P-band analysis within the main paper. This will allow for a direct and valuable comparison of the subsurface features at different frequencies.
We have also prepared a direct comparison between L- (left side) and P-band (right side) data over a smaller, representative sample area of 1km in azimuth in both HH and HV polarizations. This figure includes Pauli RGB representations on top with a white transect line indicating the tomogram locations. However, the description and interpretation of this plot is very similar to what is already presented in Section 4.3.1. Therefore, to avoid redundancy and maintain the focus and conciseness of the main manuscript, we propose not to include this plot in the paper itself.

[Figure]

Line 376-378
'The general drop of coherence with increasing…. is typical for two scattering components with a certain vertical distance.' Please add a reference.

Yes, certainly, thank you. We will add the reference of Cloude, S. R., & Papathanassiou, K. P. (2003) for clarification.

Figure 8 and 9
I miss here a brief discussion about the values of surface-to-volume ratio obtained to fit the data. Are they reasonable/explainable across polarizations (HH vs HV) and frequencies (L vs P)? The information reported in the appendix is explaining in more details the modelling approach and the obtained results, but it is not discussing them. Please, also provide bigger images.

Thank you for noting the absence of a discussion on the surface-to-volume ratio values for Figures 8 and 9. We agree that a brief discussion of these values is important for a comprehensive understanding; which will be added in Section 4.4. Overall, our modeling approach successfully fits both radar-dark and radar-bright features using the same structural model across all tested polarizations and frequencies for each feature type with the main difference being the different surface-to-volume ratios (as detailed in Tables A1 and A2).
Polarization (HH vs. HV):
- It is generally expected that HH polarization exhibits a higher surface, while HV polarization is more indicative of volume/ subsurface scattering.
- For the radar-bright features, the modeled curves show only slight changes between HH and HV, indicating a consistent scattering behavior with HH having a somewhat higher surface contribution.
- For the radar-dark features, however, the ratio difference between HH and HV is significantly more pronounced. HH polarization shows predominantly surface scattering, whereas HV polarization clearly exhibits a strong volume decorrelation over kz_vol. This results in distinctly different curve shapes between HH and HV for these features.
Frequency (L vs. P):
- Consistent with physical expectations, lower frequencies (P-band) typically penetrate deeper into the medium and are less sensitive to surface roughness, leading to a relatively reduced surface scattering component compared to volume scattering.
- For both radar-bright and radar-dark features, the overall surface-to-volume ratios for L-band are generally higher than for P-band. This trend aligns with the expectation that P-band's longer wavelength interacts more significantly with subsurface volume elements.

Section 5.1 and 5.2
In my opinion, these 2 paragraphs could be removed since they mostly summarize the findings of the analysis carried out in the previous sections.

These two paragraphs indeed summarize the key findings from Section 4. We believe their inclusion is beneficial, particularly for readers who may not be intimately familiar with all the detailed analyses presented in the preceding subsections. This direct summary serves as a bridge, before we move into the discussion, thereby facilitating a clearer understanding of our interpretation and conclusion.

Figures
Please expand Figures 1, 3, 5 to full page width and with better resolution. In some cases, it is difficult to observe the features and patterns discussed in the text (e.g. in Fig. 5).

Yes, thank you for pointing that out. We agree that improving the visibility of the figures is crucial. We will expand Figures 1, 3, and 5 to full page width and ensure they have better resolution, making it easier to observe the features and patterns discussed in the text.

---

## Author Response (AR1)

**Review: Nanna B. Karlsson**

Dear Sara-Patricia Schlenk and co-authors,

Thank you for your thorough response to the referees' comments. I am overall happy with your outlined changes and would like to invite you to submit a revised version for consideration. Based on the responses, it was not always completely clear whether you intended to implement some of the discussion points raised in your replies. I therefore iterate here for clarity where it is necessary to include your response to the referees specifically.

Dear Prof. Karlsson,

We are very grateful for your insightful comments and valuable guidance during the revision of our manuscript. We have carefully considered all the detailed feedback provided and have implemented the necessary changes throughout the text. Our point-by-point responses and manuscript adjustments are outlined following your comments below.

Please add the following to the revised manuscript:

Given that both referees suggest that data from C-band acquisitions are included, I ask you to include a couple of sentences describing that you did analyse data from this band and briefly summarise your findings, whether the analysis supports your hypothesis.

We appreciate the suggestion from both referees to include C-band data analysis. We have updated Sections 1 and 2 to confirm that C-band data was acquired during the campaign and to clarify the previous research by Pardini et al., 2016 and Parrella et al., 2021. However, we respectfully argue against including a brief summary of C-band results in this manuscript. The introduction of C-band findings would necessitate an extensive contextual discussion regarding the differing interaction mechanisms. A sentence or two would risk detracting from the focus of our primary hypothesis by raising questions that cannot be adequately addressed within the current scope.

Describe that you have investigated the effect of incidence angle and found it to be secondary to the spatial variations you observe.

We have described the influence of the incidence angle on our observations in Section 4.1.

Add information to the effect that you have looked into CRESIS OIB data and found similar subsurface features as in the sounder data.

We have incorporated information regarding the CRESIS OIB data in the tomography section (Section 4.3.1), confirming that similar subsurface features were observed in both datasets.

Add a sentence regarding the results that can be derived from the tomograms from X- and L-band.

We have inserted a sentence in the specified section (Section 4.3.1) that now clarifies the distinct subsurface information derived from the X-band and L-band tomograms.

Briefly explain why the matrix filter to suppress surface scattering contribution picks up a layer in 30 to 50m depth

We have inserted a sentence in the relevant section (Section 4.3.1) that further explains the matrix filter.

In addition, I would like to request the following:

Consider whether the new image showing backscatter for the different bands, together with a summer image, could be included in the appendix.

We have considered this suggestion and included the comparison image showing backscatter for different bands and the summer reference image in Appendix B.1.

I agree with referee Dr. Helm that H and /or HH, HV power along the flight track can easily be added to Fig. 6 without subtracting from the focus of the manuscript.

We have considered the suggestion and added the Entropy (H) along the flight track as Figure 6f, addressing the request to include additional polarimetric information.

I also agree that figures should be colourblind friendly. I appreciate that there is a standard presentation of tomography in the community; however, a quick Google search returned many examples of tomography plots using the viridis colour palette, so I don't think your argument is valid here.

We acknowledge the point regarding the use of community standards versus colorblind-friendly palettes. We have revised the figures and now use the Viridis color palette for all tomography plots to ensure accessibility, as suggested.

Please clarify the following response: "This 'band' is generally located deeper for the radar-dark features and higher for the radar-dark features (see Tomographic analysis and Modelling approach)." Which is radar-dark and which is radar-bright?

Thank you for noting this error. We apologize for the ambiguity and have clarified the text below to correctly distinguish between the radar-dark and radar-bright features.
The "dark reflection band" you describe aligns with what we interpret as our subsurface scattering layer, representing the transition between the low-backscatter layer above and the higher-backscatter area below. This "band" is generally located deeper for the radar-dark features and higher for the **radar-bright** features (see Tomographic analysis and Modelling approach).